# How cells tame noise while maintaining ultrasensitive transcriptional responses

Eui Min Jeong[1,2☯*], Chang Yoon Chung[2,3☯], Jae Kyoung Kim [2,4,5,6¤*]

1 Department of Data Science, Inha University, Incheon, Republic of Korea, 2 Biomedical Mathematics Group, Pioneer Research Center for Mathematical and Computational Sciences, Institute for Basic Science, Daejeon, Republic of Korea, 3 Department of Mathematics and Statistics, Université de Montréal, Montréal, Québec, Canada, 4 Department of Mathematical Sciences, KAIST, Daejeon, Republic of Korea, 5 Department of Medicine, College of Medicine, Korea University, Seoul, Republic of Korea, 6 Graduate School of Data Science, KAIST, Daejeon, Republic of Korea

☯ These authors contributed equally to this work.
¤ Current Address: Department of Mathematical Sciences, KAIST, 291, Daehak-ro Yuseong-gu, Daejeon Republic of Korea
* wjddmlals11@inha.ac.kr, jaekkim@kaist.ac.kr

## Abstract

Ultrasensitive transcriptional switches are essential for converting gradual molecular inputs into decisive gene expression responses, enabling critical behaviors such as bistability and oscillations. While cooperative binding, relying on direct repressor-DNA binding, has been classically regarded as a key ultrasensitivity mechanism, recent theoretical works have demonstrated that combinations of indirect repression mechanisms—sequestration, blocking, and displacement—can also achieve ultrasensitive switches with greater robustness to transcriptional noise. However, these previous works have neglected key biological constraints such as DNA binding kinetics and the limited availability of transcriptional activators, raising the question of whether ultrasensitivity and noise robustness can be sustained under biologically realistic conditions. Here, we systematically assess the impact of these factors on ultrasensitivity and noise robustness under physiologically plausible conditions. We show that while various repression combinations can reduce noise, only the full combination of all three indirect mechanisms consistently maintains low noise and high ultrasensitivity. As a result, biological oscillators employing this triple repression architecture retain precise rhythmic switching even under high noise, and even when activators are shared across thousands of target genes. Our findings offer a mechanistic explanation for the frequent co-occurrence of these repression mechanisms in natural gene regulatory systems.

**Data availability statement:** All relevant code for data generation and analysis in this study is available at https://github.com/Mathbiomed/Ultrasensitive-gene-switch.

**Funding:** The research was supported by the following agencies and institutions: the Institute for Basic Science (grant no. IBS-R029-C3, J.K.K.); Samsung Science and Technology Foundation (grant no. SSTF-BA1902-01, J.K.K.); the National Research Foundation of Korea (NRF) grant funded by the Korean government (MSIT) (grant no. RS-2022-NR068758, J.K.K.); Basic Science Research Program through the National Research Foundation of Korea (NRF) funded by the Ministry of Education (grant no. RS-2025-25397599, J.K.K.); INHA UNIVEERSITY Research Grant (E.M.J.). The funders had no role in study design, data collection and analysis, decision to publish, or preparation of the manuscript.

## Author summary

Cells must make accurate decisions in noisy environments using limited molecular resources. One essential tool for this is the ultrasensitive transcriptional switch, which enables sharp transitions in gene expression. While cooperative binding has long been viewed as the primary mechanism behind ultrasensitivity, it is highly sensitive to molecular noise. Our study explores how cells overcome this limitation by combining three indirect repression mechanisms: sequestration, blocking, and displacement. We show that this triple-repression architecture not only generates ultrasensitivity but also ensures noise robustness under physiologically realistic conditions—even when a single pool of transcription factors regulates thousands of genes. These findings reveal a biologically feasible strategy for noise-resilient transcription and offer a mechanistic explanation for why these repression strategies frequently co-occur in natural systems.

## Introduction

Ultrasensitivity, characterized by a sharp output change in response to a small variation in input, underlies many essential regulatory functions in biological systems, including signal amplification, bistability, and oscillatory dynamics [1–4]. One of the most well-known mechanisms that produce ultrasensitive responses is cooperative binding, which is broadly observed across biology, in phenomena such as receptor-ligand binding [5,6], protein–RNA interactions [7], and small-molecule binding [8]. A prominent example in transcriptional regulation is the cooperative binding of repressors to multiple DNA sites, which enables the formation of sharp transcriptional switches, where small changes in repressor concentration lead to abrupt transcriptional repression [9,10]. While a cooperative binding mechanism has traditionally been considered a primary strategy for achieving ultrasensitivity, recent theoretical studies have highlighted alternative approaches [11–20]. In particular, combinations of indirect repression mechanisms—such as sequestration, blocking, and displacement—have been shown through ordinary differential equation (ODE) models to produce comparably strong ultrasensitive responses without requiring direct DNA binding [21]. Building on this, stochastic modeling studies further revealed that unlike cooperative binding, which is susceptible to transcriptional noise, systems employing multiple indirect repression mechanisms exhibit enhanced noise robustness [22]. Collectively, these findings suggest that combinations of indirect repression mechanisms can generate ultrasensitive transcriptional switches that are more robust to noise than a cooperative binding mechanism.

However, these previous studies typically assumed fixed or idealized conditions, without explicitly accounting for key biological parameters such as DNA binding/unbinding rates and the abundance of transcriptional activators [21,22]. In living cells, the kinetics of molecular interactions and the limited availability of regulatory proteins impose significant constraints on transcriptional dynamics [23–25]. It remains unclear

whether ultrasensitivity and noise robustness can still be achieved under such physiologically realistic conditions. More-over, it is unknown which combinations of repression mechanisms, if any, are capable of maintaining performance under these constraints. Addressing this gap is crucial for understanding how biological systems maintain both precision and sensitivity in gene regulation despite molecular noise and finite resources.

To address this, we explicitly incorporated DNA binding rates and activator copy numbers into the models and system-atically evaluated how these biologically constrained parameters impact ultrasensitivity and noise robustness. Our results show that, regardless of the specific repression mechanism, indirect repression can reduce noise when DNA binding is fast and activators are abundant, indicating that previous conclusions [22] were incomplete. However, only the full com-bination of all three mechanisms consistently achieves low noise levels and high ultrasensitivity within physiologically plausible constraints. Accordingly, biological oscillators employing this triple repression architecture can generate precise rhythms, reliably toggling the transcriptional switch on and off even in the presence of stochastic fluctuations. We further demonstrate that this robustness is preserved even when a single pool of activators simultaneously regulates thousands of target genes rather than a single gene, highlighting the scalability and efficiency of this architecture under conditions mimicking a natural gene regulatory network. These findings offer a mechanistic rationale for the frequent co-occurrence of sequestration, blocking, and displacement in natural transcriptional circuits and present a design principle for construct-ing resource-efficient and noise-resilient gene regulatory systems.

## Results

### Ultrasensitivity generated with cooperative binding is noisy

The cooperative binding mechanism, in which the transcriptional repressors bind cooperatively to multiple DNA sites to inhibit transcription, is one of the most common transcriptional mechanisms for generating ultrasensitivity [5,9–11,26]. To achieve an ultrasensitive transcriptional response, we used a model describing the cooperative binding with four independent binding sites (Fig 1a). In the model, free DNA ($E_{0000}$) contains four binding sites, where repressors ($R$) bind at a rate of $k_f$ and unbind with rates of $k_r$, $ck_r$, $c^2k_r$, and $c^3k_r$, for cases in which one ($E_{0001}$, $E_{0010}$, $E_{0100}$, and $E_{1000}$), two ($E_{1100}$, $E_{1010}$, $E_{1001}$, $E_{0110}$, $E_{0101}$, and $E_{0011}$), three ($E_{1110}$, $E_{1101}$, $E_{1011}$, and $E_{0111}$), and all four ($E_{1111}$) sites are occupied by $R$, respectively. When all sites are occupied, transcription is inhibited, but it remains active when at least one site is unoccupied, producing mRNA at a rate of $\alpha$, which degrades at a rate of $\beta$ (Table 1). Thus, the transcriptional activity (i.e., the probability at which the transcription is active), derived using the chemical master equation (CME) framework (see Methods), decreases as the number of repressors ($R_T$) increases. Notably, the transcriptional activity can be changed sensitively with respect to changes in $R_T$ in the presence of cooperativity (i.e., $c < 1$), achieving ultrasensitivity comparable to the Hill exponent of four (Fig 1b).

This ultrasensitivity is maintained as long as the dissociation constant $K_r = k_r/k_f$ remains fixed, even when $k_f$ varies. However, the noise level changes when $k_f$ varies. For instance, as $k_f = 6 \times 10^2$ is increased to $k_f = 6 \times 10^5$, while keeping $K_r = 10^{-2}$, the level of ultrasensitivity does not change (Fig 1b), but the noise level quantified with the Fano factor (i.e., variance/mean) of the mRNA copy numbers considerably decreases (Fig 1c). To investigate how increasing the values of $k_f$ and $k_r$ attenuates transcriptional noise, we simulated the stationary distribution of mRNA copy numbers (Fig 1d) using the Gillespie algorithm (see Methods) [27]. When $k_f$ and $k_r$ became slow relative to the rates of mRNA production and degradation (i.e., $\alpha$ and $\beta$), slow transitions between the active and repressed DNA states resulted in bimodal mRNA sta-tionary distributions (Fig 1d, top). Such noise attenuation by fast $k_f$ and $k_r$ was also observed in an activator model, where transcription occurs only when an activator protein binds cooperatively to all four sites (S1 Fig), consistent with Sánchez et al [28].

Next, we investigate whether the noise level in transcription decreases as the binding and unbinding rates become faster across a broad parameter range. For this, while keeping $K_r = 10^{-2}$ and $\alpha/\beta = 100$, we varied $k_f$ and $\beta$. We found

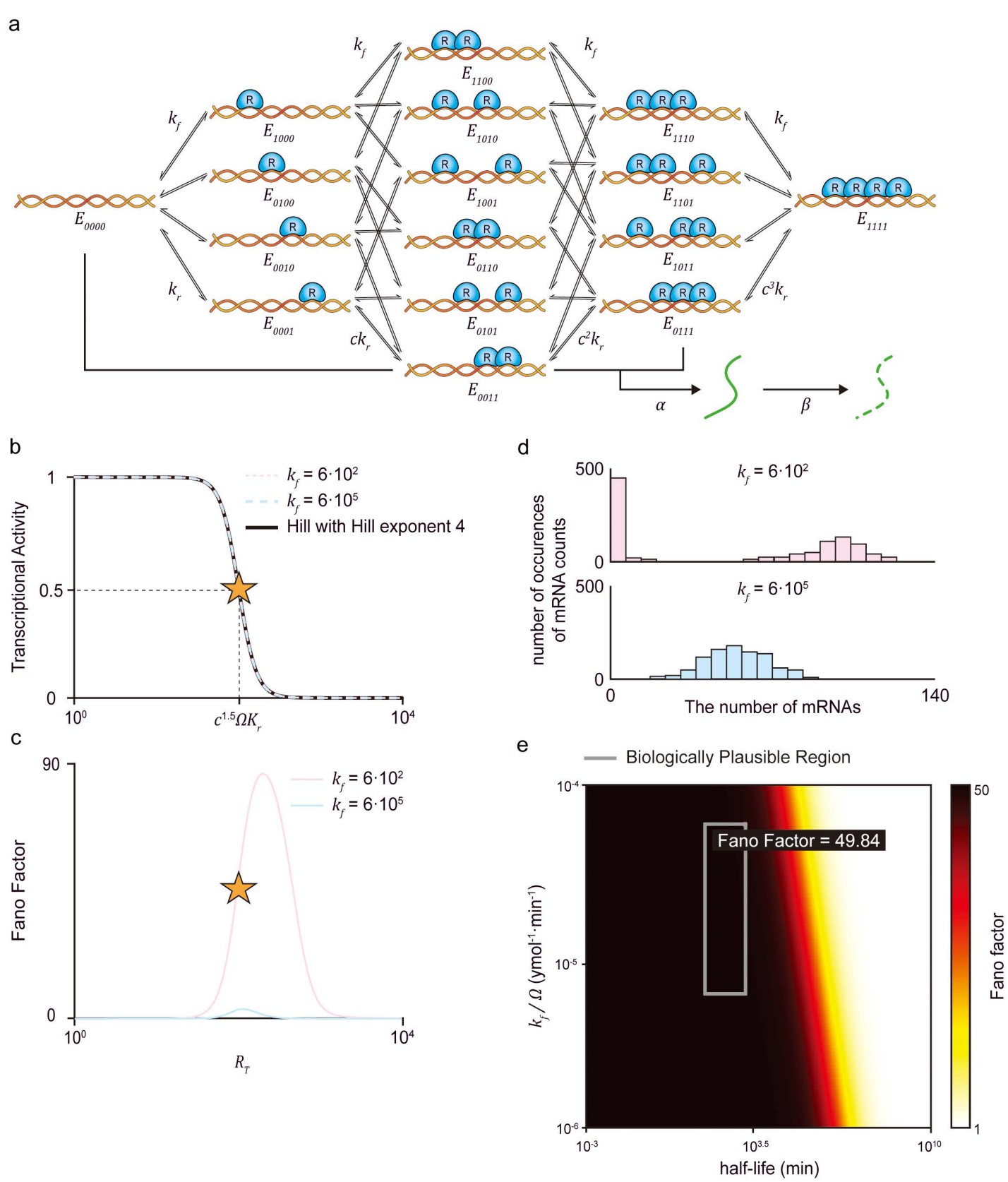

PLOS Computational Biology

**Fig 1. The cooperative binding mechanism is sensitive to noise under biologically realistic conditions.** (a) Model diagram of the transcription regulated by the repressor proteins ($R$) binding to four independent sites on the DNA within a cell volume of $\Omega$. Each binding site is occupied by $R$ at a rate $k_f$. Conversely, $R$ unbinds from DNA at a rate $k_r$ when one site is occupied, with the dissociation constant between $R$ and DNA defined as $K_r = k_r/k_f$. For two, three, or four occupied sites, $R$ unbinds at rates of $ck_r$, $c^2k_r$, and $c^3k_r$, respectively. Accordingly, when $c < 1$, cooperative binding is present. When all binding sites are occupied, transcription is inhibited. On the other hand, when any sites remain unoccupied, mRNA is produced at a rate $\alpha$ and degrades at a rate $\beta$. (b) When $c = 10^{-2}$, transcriptional activity closely resembles the Hill function with Hill exponent of 4 (black line). Furthermore, transcriptional activity remains consistent with the Hill function regardless of $k_f$ values, provided $K_r$ is kept constant (blue dashed line and red dashed line). (c) Nevertheless, the overall noise level quantified by the Fano factor of mRNAs shows significant differences based on $k_f$. Such differences are particularly evident when transcriptional activity undergoes sensitive response. Notably, the overall noise levels are reduced during the sensitive response when $k_f$ is faster (blue solid line) compared to when it is slower (red solid line). (d) To investigate how differences in $k_f$ values affect noise levels, the stationary distributions of mRNAs are simulated at the transcriptional activity of 0.5, where transcriptional activity exhibits the most sensitive response (b and c, star). A slower $k_f$ results in DNA dynamics that are slower relative to mRNA dynamics (i.e., $k_f\Omega^{-1}$, $k_r \ll \alpha, \beta$), causing slow transitions between inhibited and activated DNA complexes. These slow transitions between inhibited and activated DNA complexes lead to a bimodal distribution of mRNAs that significantly deviates from the Poisson distribution, leading to a high Fano factor (top). On the other hand, a faster $k_f$ produces unimodal distributions that resemble a Poisson distribution, causing a reduced Fano factor (bottom). (e) Since the relative rates of DNA dynamics (i.e., $k_f\Omega^{-1}$ and $k_r$) and mRNA dynamics (i.e., $\alpha$ and $\beta$) affect noise levels, the Fano factors of mRNA distributions are examined under varying rates. Specifically, the Fano factor at the transcriptional activity of 0.5 is calculated with respect to $k_f$ and the half-life of mRNA ($\ln 2/\beta$), while maintaining the dissociation constant $K_r = k_r/k_f$ and effective transcription rate $\alpha/\beta$. When $k_f$ or the half-life is higher and thus $k_f$ becomes faster relative to mRNA dynamics, the Fano factor decreases. However, with biologically realistic $k_f$ and half-life (gray box), noise levels remain substantially high (Fano factor > 45).

that, when the binding rate is sufficiently fast relative to the mRNA half-life ($\ln 2/\beta$), the Fano factor approached its minimal value of 1 (Fig 1e). However, achieving this requires biologically unrealistic fast binding and unbinding rates. Specifically, with a biologically relevant binding rate $k_f\Omega^{-1} = 6 \times 10^{-6} \sim 6 \times 10^{-5}$ $ymol^{-1}min^{-1}$ [24,29,30], where $\Omega = 10^7 aL$ is the volume of the cell [23], and the mRNA half-life is 0.5~16.4 hours (i.e., 30~984 min; Table 2) [31,32], the Fano factor is greater than 40 (Fig 1e, gray box). Taken together, while increasing binding rates can minimize noise, the cooperative binding mechanism remains highly noisy under physiological conditions.

## Ultrasensitivity generated with indirect repression mechanisms is robust to noise

The transcriptional repression by repressors can occur not only through direct binding to DNA such as cooperative binding mechanism, but also by indirectly inhibiting activators that promote transcription [14,33–36]. In addition, combining multiple indirect repression mechanisms can generate an ultrasensitive transcriptional response [21], while maintaining lower noise levels [22]. Accordingly, we investigated whether the models describing different combinations of indirect repression mechanisms can achieve ultrasensitivity with low transcriptional noise under biologically plausible conditions.

In this model, activators ($A$) bind to free DNA ($E_F$) at a rate $k_f$ to form activator-bound DNA ($E_A$), and unbind at a rate $k_a$. Once $E_A$ is formed, the transcription is activated, leading to mRNAs production at a rate $\alpha$ and degradation at a rate $\beta$ (Fig 2a, bottom). To inhibit the transcription, indirect repression mechanisms such as sequestration, blocking, and displacement can be used. Specifically, free repressors ($R$) can bind to $A$ at a rate $k_f$, forming an activator-repressor complex ($R_A$) to prevent $A$ from binding to DNA (sequestration; Fig 2a, top left), while unbinding from $R_A$ at a rate $k_s$. $R$ can also bind directly to $E_A$ at a rate $k_f$, forming the activator-repressor-bound DNA ($E_R$), thereby blocking transcription (blocking; Fig 2a, top middle), while $R$ can unbind from $E_R$ at a rate $k_b$. Additionally, $R$ removes $A$ from $E_R$ with the form of $R_A$ at a rate $k_d$, whereas $R_A$ can bind to DNA at a rate $k_f$ reversely (displacement; Fig 2a, top right, and Table 1). Based on these reactions, we constructed three distinct models one with sole sequestration by including only $A$, $R$, and $R_A$ dynamics; a second with sequestration and blocking by additionally incorporating $R$ binding/unbinding to $E_A$; and a third model with all three repressions by further including binding/unbinding between $R_A$ and $E_F$.

First, by adjusting the dissociation constants $K_s = k_s/k_f$ for sequestration, the model with sequestration alone can achieve ultrasensitivity comparable to that of cooperative binding with four binding sites (Fig 2b, red solid line). Subsequently, adjusting $K_b = k_b/k_f$ for blocking and $K_d = k_d/k_f$ for displacement enables other two models to reach similar levels of ultrasensitivity (Fig 2b, blue and yellow dashed lines). While the level of their ultrasensitivity is similar, interestingly,

**Table 1. Propensity functions of reactions and parameter values for all models.**

| Model | Reaction | Propensity function | Parameter value |
|---|---|---|---|
| The four binding sites model | $E_X \to E_X + M$<br>$X \in \{0000, 0001, 0010,$<br>$0100, 1000, 0011, 0101,$<br>$1001, 0110, 1010, 1100,$<br>$0111, 1011, 1101, 1110\}$ | $\alpha n_{E_X}$ | $\alpha = 100\beta,$<br>$\beta = \frac{\ln 2}{984.5}\ min^{-1},$<br>$c = 10^{-2},$<br>$k_f = 600,\ 6 \cdot 10^5\ aL$<br>$\quad \cdot ymol^{-1} min^{-1},$<br>$\Omega = 10^7\ aL,$<br>$K_r = 10^{-2}\ ymol$<br>were used in Fig 1b–1c.<br>$R_T = c^{1.5}\Omega K_r = 100$<br>was used in Fig 1d. |
| | $M \to \phi$ | $\beta n_M$ | |
| | $E_{0000} \to E_X$<br>$X \in \{0001, 0010,$<br>$0100, 1000\}$ | $(k_f/\Omega)\, R_T n_{E_{0000}}$ | |
| | $E_{0001} \to E_X$<br>$X \in \{0011, 0101, 1001\}$ | $(k_f/\Omega)\, R_T n_{E_{0001}}$ | |
| | $E_{0010} \to E_X$<br>$X \in \{0011, 0101, 1001\}$ | $(k_f/\Omega)\, R_T n_{E_{0010}}$ | |
| | $E_{0100} \to E_X$<br>$X \in \{0101, 0110, 1100\}$ | $(k_f/\Omega)\, R_T n_{E_{0100}}$ | |
| | $E_{1000} \to E_X$<br>$X \in \{1001, 1010, 1100\}$ | $(k_f/\Omega)\, R_T n_{E_{1000}}$ | |
| | $E_{0011} \to E_X$<br>$X \in \{0111, 1011\}$ | $(k_f/\Omega)\, R_T n_{E_{0011}}$ | |
| | $E_{0101} \to E_X$<br>$X \in \{0111, 1101\}$ | $(k_f/\Omega)\, R_T n_{E_{0101}}$ | |
| | $E_{1001} \to E_X$<br>$X \in \{1011, 1101\}$ | $(k_f/\Omega)\, R_T n_{E_{1001}}$ | |
| | $E_{0110} \to E_X$<br>$X \in \{0111, 1110\}$ | $(k_f/\Omega)\, R_T n_{E_{0110}}$ | |
| | $E_{1010} \to E_X$<br>$X \in \{1011, 1110\}$ | $(k_f/\Omega)\, R_T n_{E_{1010}}$ | |
| | $E_{1100} \to E_X$<br>$X \in \{1101, 1110\}$ | $(k_f/\Omega)\, R_T n_{E_{1100}}$ | |
| | $E_{0111} \to E_{1111}$ | $(k_f/\Omega)\, R_T n_{E_{0111}}$ | |
| | $E_{1011} \to E_{1111}$ | $(k_f/\Omega)\, R_T n_{E_{1011}}$ | |
| | $E_{1101} \to E_{1111}$ | $(k_f/\Omega)\, R_T n_{E_{1101}}$ | |
| | $E_{1110} \to E_{1111}$ | $(k_f/\Omega)\, R_T n_{E_{1110}}$ | |
| | $E_X \to E_{0000}$<br>$X \in \{0001, 0010,$<br>$0100, 1000\}$ | $k_r n_{E_X}$ | |
| | $E_X \to E_{0001}$<br>$X \in \{0011, 0101, 1001\}$ | $c k_r n_{E_X}$ | |
| | $E_X \to E_{0010}$<br>$X \in \{0011, 0101, 1001\}$ | $c k_r n_{E_X}$ | |
| | $E_X \to E_{0100}$<br>$X \in \{0101, 0110, 1100\}$ | $c k_r n_{E_X}$ | |
| | $E_X \to E_{1000}$<br>$X \in \{1001, 1010, 1100\}$ | $c k_r n_{E_X}$ | |
| | $E_X \to E_{0011}$<br>$X \in \{0111, 1011\}$ | $c^2 k_r n_{E_X}$ | |
| | $E_X \to E_{0101}$<br>$X \in \{0111, 1101\}$ | $c^2 k_r n_{E_X}$ | |
| | $E_X \to E_{1001}$<br>$X \in \{1011, 1101\}$ | $c^2 k_r n_{E_X}$ | |

| Model | | | Reaction | Propensity function | Parameter value |
|---|---|---|---|---|---|
| | | | $E_X \to E_{0110}$ <br> $X \in \{0111, 1110\}$ | $c^2 k_r n_{E_X}$ | |
| | | | $E_X \to E_{1010}$ <br> $X \in \{1011, 1110\}$ | $c^2 k_r n_{E_X}$ | |
| | | | $E_X \to E_{1100}$ <br> $X \in \{1101, 1110\}$ | $c^2 k_r n_{E_X}$ | |
| | | | $E_{1111} \to E_{0111}$ | $c^3 k_r n_{E_X}$ | |
| | | | $E_{1111} \to E_{1011}$ | $c^3 k_r n_{E_X}$ | |
| | | | $E_{1111} \to E_{1101}$ | $c^3 k_r n_{E_X}$ | |
| | | | $E_{1111} \to E_{1110}$ | $c^3 k_r n_{E_X}$ | |
| The combination of indirect repressions model | with the sole sequestration | | $E_A \to E_A + M$ | $\alpha n_{E_A}$ | $\alpha = 100\beta,$ <br> $\beta = \frac{\ln 2}{984.5}\ min^{-1},$ <br> $k_f = 600\ aL$ <br> $\cdot ymol^{-1} min^{-1},$ <br> $\Omega = 10^7\ aL,$ <br> $A_T = 10^3, 10^4, 10^5,$ <br> $\frac{K_s}{A_T} = 4.5 \cdot 10^{-10}\ ymol,$ <br> $\frac{K_a}{A_T} = 7 \cdot 10^{-9}\ ymol,$ <br> $\frac{K_b}{A_T} = 10^{-6}\ ymol,$ <br> $K_d = K_a \cdot K_b / K_s$ <br> were used in Fig 2b–2c. <br> $\frac{K_s}{A_T} = 6 \cdot 10^{-13}\ ymol,$ <br> $\frac{K_a}{A_T} = 6 \cdot 10^{-11}\ ymol,$ <br> $\frac{K_b}{A_T} = 10^{-6}\ ymol,$ <br> $K_d = K_a \cdot K_b / K_s$ <br> were used in Fig 3a–3b. |
| | | | $M \to \phi$ | $\beta n_M$ | |
| | | | $A + E_F \to E_A$ | $(k_f/\Omega)\, A\,(R_T, A_T, K_s)\, n_{E_F}$ | |
| | | | $E_A \to A + E_F$ | $k_a n_{E_A}$ | |
| | with the blocking | | $R + E_A \to E_R$ | $(k_f/\Omega)\, R\,(R_T, A_T, K_s)\, n_{E_A}$ | |
| | | | $E_R \to R + E_A$ | $k_b n_{E_R}$ | |
| | with the displacement | | $R_A + E_F \to E_R$ | $(k_f/\Omega)\, R_A\,(R_T, A_T, K_s)\, n_{E_F}$ | |
| | | | $E_R \to R_A + E_F$ | $k_d n_{E_R}$ | |
| The transcriptional NFL model | with the sole sequestration | | $E_A \to E_A + M$ | $\alpha n_{E_A}$ | $\alpha = 100\beta,$ <br> $\beta = \frac{\ln 2}{120}\ min^{-1},$ <br> $\alpha_2 = 5/12\ min^{-1},$ <br> $\alpha_3 = 5/12\ min^{-1},$ <br> $k_f = 600\ aL$ <br> $\cdot ymol^{-1} min^{-1},$ <br> $\frac{K_s}{A_T} = 9 \cdot 10^{-12}\ ymol,$ <br> $\frac{K_a}{A_T} = 2 \cdot 10^{-9}\ ymol,$ <br> $\frac{K_b}{A_T} = 5 \cdot 10^{-9}\ ymol,$ <br> $K_d = K_a \cdot K_b / K_s$ <br> $\Omega = 10^7\ aL,$ <br> $A_T = 10^5,$ <br> were used in Fig 4g. |
| | | | $M \to \phi$ | $\beta n_M$ | |
| | | | $M \to R_c$ | $\alpha_2 n_M$ | |
| | | | $R_c \to \phi$ | $\beta n_{R_c}$ | |
| | | | $R_c \to R$ | $\alpha n_{E_A}$ | |
| | | | $R \to \phi$ | $\beta n_R$ | |
| | | | $A + E_F \to E_A$ | $(k_f/\Omega)\, A\,(R_T, A_T, K_s)\, n_{E_F}$ | |
| | | | $E_A \to A + E_F$ | $k_a n_{E_A}$ | |
| | with the blocking | | $R + E_A \to E_R$ | $(k_f/\Omega)\, R\,(R_T, A_T, K_s)\, n_{E_A}$ | |
| | | | $E_R \to R + E_A$ | $k_b n_{E_R}$ | |
| | with the displacement | | $R_A + E_F \to E_R$ | $(k_f/\Omega)\, R_A\,(R_T, A_T, K_s)\, n_{E_F}$ | |
| | | | $E_R \to R_A + E_F$ | $k_d n_{E_R}$ | |

*(Continued)*

| Model | | Reaction | Propensity function | Parameter value |
|---|---|---|---|---|
| The multi-target gene regulation model | with the sole sequestration | $E_A \rightarrow E_A + M$ | $\alpha n_{E_A}$ | $\alpha = 100\beta$, $\beta = \frac{\ln 2}{120} \ min^{-1}$, $k_f = 600 \ aL$ $\cdot ymol^{-1} min^{-1}$, $K_s = 4.5 \cdot 10^{-10} \ ymol$ $K_a = 7 \cdot 10^{-9} \ ymol$ $K_b = 10^{-6} \ ymol$ $K_d = K_a \cdot K_b / K_s$ $\Omega = 10^7 \ aL$, $A_T = 10^3$, $D_T = 0, \ 10, \ 100,$ $1,000, \ 2,000,$ were used in Fig 5b–5d. |
| | | $M \rightarrow \phi$ | $\beta n_M$ | |
| | | $A + E_F \rightarrow E_A$ | $(k_f/\Omega) \, n_A n_{E_F}$ | |
| | | $E_A \rightarrow A + E_F$ | $k_a n_{E_A}$ | |
| | | $A + D_F \rightarrow D_A$ | $(k_f/\Omega) \, n_A n_{D_F}$ | |
| | | $D_A \rightarrow A + D_F$ | $k_a n_{D_A}$ | |
| | | $R + A \rightarrow R_A$ | $(k_f/\Omega) \, n_R n_A$ | |
| | | $R_A \rightarrow R + A$ | $k_s n_{R_A}$ | |
| | with the blocking | $R + E_A \rightarrow E_R$ | $(k_f/\Omega) \, n_R n_{E_A}$ | |
| | | $E_R \rightarrow R + E_A$ | $k_b n_{E_R}$ | |
| | | $R + D_A \rightarrow D_R$ | $(k_f/\Omega) \, n_R n_{D_A}$ | |
| | | $D_R \rightarrow R + D_A$ | $k_b n_{D_R}$ | |
| | with the displacement | $R_A + E_F \rightarrow E_R$ | $(k_f/\Omega) \, n_{R_A} n_{E_F}$ | |
| | | $E_R \rightarrow R_A + E_F$ | $k_d n_{E_R}$ | |
| | | $R_A + D_F \rightarrow D_R$ | $(k_f/\Omega) \, n_{R_A} n_{D_F}$ | |
| | | $D_R \rightarrow R_A + D_F$ | $k_d n_{D_R}$ | |

Here, $n_X$ is the number of $X$; $A_T$, $R_T$, and $D_T$ are the total number of activators, repressors, and additional target genes, respectively. Under the assumption that the binding/unbinding reactions between proteins (i.e., $R$ and $A$) equilibrate much faster than those between proteins and DNA (i.e., $A$ and $E_F$, $R$ and $E_A$, and $R_A$ and $E_F$), the number of proteins (i.e., $R$, $A$, and $R_A$) equilibrate rapidly and can be approximated by their quasi-steady-state approximations (QSSAs). These are given by:

$A(R_T, A_T, K_s) = \frac{A_T - R_T - \Omega K_s + \sqrt{(A_T - R_T - \Omega K_s)^2 + 4\Omega A_T K_s}}{2}$, $R(R_T, A_T, K_s) = \frac{R_T - A_T - \Omega K_s + \sqrt{(A_T - R_T - \Omega K_s)^2 + 4\Omega A_T K_s}}{2}$, and $R_A(R_T, A_T, K_s) = \frac{A_T + R_T + \Omega K_s - \sqrt{(A_T - R_T - \Omega K_s)^2 + 4\Omega A_T K_s}}{2}$.
This QSSA assumption has been validated through stochastic simulations (S2 Fig).

**Table 2. The references of parameter values used for model simulations.**

| Cell type | Parameter (definition) | Reported range | References |
|---|---|---|---|
| In vitro (Cell host: human NF-κB and IκBα expressed in E. coli BL21 DE3); N/A – Theoretical/ literature-based study; In silico molecular dynamics simulation | $k_f$ (binding rate) | $10^6 \sim 10^7 \ M^{-1} s^{-1}$ | Williamson, 2023; Bergqvist et al., 2009; Zou et al., 2020 |
| Typical mammalian cells | $\Omega$ (cell volume) | $10^{-12} \sim 10^{-11} l$ | Milo et al., 2016 |
| Mouse embryonic stem cells, human whole blood | ln 2/ $\beta$ (mRNA half-life) | 0.5 (less than 1 h) $\sim$ 16.4 h | Sharova et al., 2009; Wang & Liu, 2022 |
| H. sapiens cell lines | $A_T/\Omega$ (transcription factor concentration) | $10^{-9} \sim 10^{-7} M$ | Milo et al., 2016 |

their noise levels differ (Fig 2c and Table 3). Specifically, as more repression mechanisms are added, the overall noise levels decrease, consistent with a previous study [22].

Next, we investigate whether the noise levels can be further decreased as binding and unbinding rates become faster as shown in cooperative binding. Indeed, as binding rates become faster relative to mRNA half-life, the Fano factor decreases in all three models. However, within the biologically relevant range of binding rate and mRNA half-life, the Fano factor still remains high, with a minimum of about 4 (Table 2 and Fig 2d, top row). In this case, we used the number of activators $A_T = 10^3$. Since the biologically relevant copy number of activators is typically within the range of $A_T = 10^3 \sim 10^5$

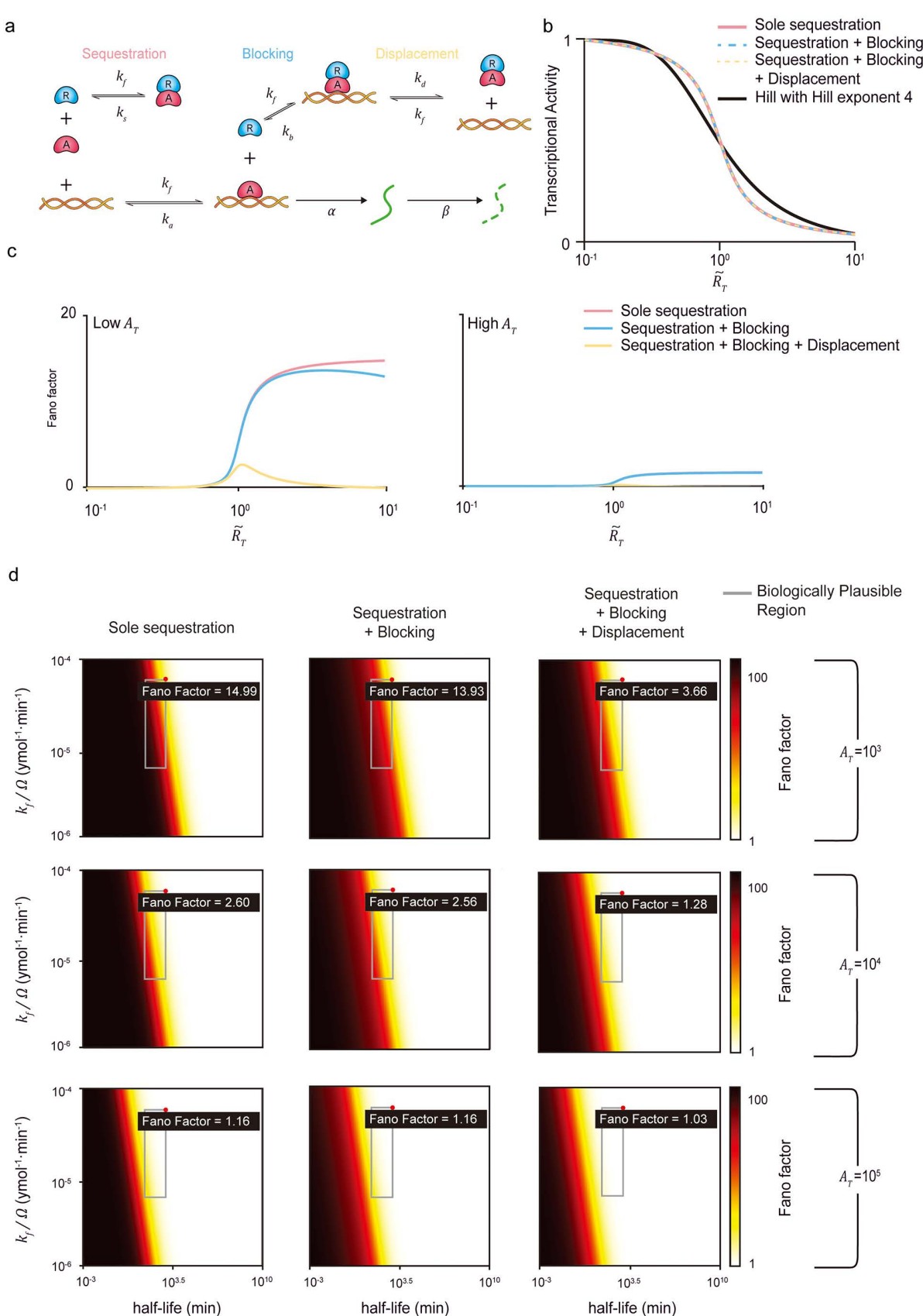

**Fig 2. The combinations of repression mechanisms can achieve minimal noise levels with high sensitivity under biologically realistic conditions.** (a) Model diagram of transcription regulated by multiple repression mechanisms. An activator ($A$) binds to DNA, forming an activated DNA complex ($E_A$) at a rate $k_f$, and unbinds from DNA at a rate $k_a$. Once $E_A$ is formed, mRNAs are produced at a rate $\alpha$ and degrade at a rate $\beta$. To prevent the formation of $E_A$, a repressor ($R$) sequesters free $A$ at a rate $k_f$, forming a repressor-activator complex ($R_A$; sequestration). $R$ also unbinds from $R_A$ at a rate $k_s$. When $A$ is already bound to DNA, a free $R$ binds directly to the DNA-bound $A$ at a rate $k_f$ forming a repressed DNA complex ($E_R$; blocking), and unbinds from $E_R$ at a rate $k_b$. Additionally, free $R$ can displace DNA-bound $A$ from $E_R$ by forming $R_A$ (displacement) at a rate $k_d$, while $R_A$ binds to DNA at a rate $k_f$. Accordingly, the dissociation constants are defined as $K_a = k_a/k_f$, $K_s = k_s/k_f$, $K_b = k_b/k_f$, and $K_d = k_d/k_f$. (b) To investigate whether the transcriptional activity of multiple repression mechanisms can exhibit a sensitive response similar to the Hill function, transcriptional activities of the sole sequestration (red solid line), the combination of sequestration and blocking (blue dashed line), and the combination of sequestration, blocking, and displacement (yellow dashed line) are derived. For all three models, when the number of repressors ($R_T$) is less than the number of activators ($A_T$), unsequestered activators can bind to DNA, promoting transcription (activation phase). However, when $R_T$ exceeds $A_T$, most activators are sequestered by repressors, thereby suppressing transcription (repression phase). In the switching from activation to repression phase, a sharp transition occurs when the molar ratio between $A_T$ and $R_T$ ($\widetilde{R_T} = R_T/A_T$) is near one. The sensitivity of this transition is determined by the dissociation constants $K_a$, $K_s$, $K_b$, and $K_d$; hence, they are appropriately adjusted to match all models that exhibit transcriptional activity comparable to the Hill function with Hill exponent of 4 (black solid line). (c) Despite this similarity in transcriptional activity, the three models show significant differences in overall noise levels. Specifically, overall noise levels decrease with the addition of repression mechanisms: from the sole sequestration (red solid line) to the combination of sequestration and blocking (blue solid line), and further the combination of sequestration, blocking, and displacement (yellow solid line). Additionally, overall noise levels across all $\widetilde{R_T}$ values are lower when $A_T$ is higher (right) compared to when it is lower (left). (d) Given the impact of $A_T$ on overall noise levels across the three models, the Fano factor of mRNAs is examined under varying $A_T$ values. Specifically, the maximum Fano factor near $R_T$ of 1 is calculated with respect to $k_f\Omega^{-1}$ and the half-life of mRNA, while maintaining the dissociation constant $K_a$, $K_b$ and $K_d$, and effective transcription rate $\alpha/\beta$. Heatmaps of the maximum Fano factor are shown for the sole sequestration (first column), the combination of sequestration and blocking (second column), and the combination of sequestration, blocking and displacement (third column). Each model is simulated with $A_T$ values of $10^3$, $10^4$, and $10^5$, all of which are biologically realistic. When $A_T = 10^3$ (first row), all three models show lower noise levels compared to the cooperative binding model over the same range of $k_f\Omega^{-1}$ and the half-life of mRNA. Furthermore, overall noise levels progressively decrease as the repression mechanisms are added. In particular, within the biologically realistic $k_f\Omega^{-1}$ and the mRNA half-life (gray box), the combination of all three mechanisms reduces the noise level down to the Fano factor of approximately 3, much lower compared to the cooperative binding (i.e., Fano factor = 49). The noise levels are further decreased as the level of $A_T$ increases. When $A_T = 10^5$ (third row), the minimum noise levels (i.e., a Fano factor of 1) are achievable within the biologically realistic $k_f$ and the mRNA half-life (gray box) for all three mechanisms.

yoctomole [23], we next tested whether increasing $A_T$ could further suppress noise. Indeed, higher $A_T$ shifted the low noise region towards the biologically realistic parameter space without altering $k_f$ or the mRNA half-life (Fig 2d, middle row). Notably, when $A_T$ was increased to $10^5$, in combination with faster binding rates and longer mRNA half-lives, the Fano factor approached ~1 in all three models (Fig 2d, bottom row).

To further investigate noise levels under conditions of higher ultrasensitivity, we adjusted the dissociation constants of the indirect repression mechanisms to produce transcriptional activity with a Hill coefficient of 50 (Fig 3a). This elevated ultrasensitivity is accompanied by increased transcriptional noise (Figs 2c and 3b). This increase amplifies the differences among the models, which originate from stochastic transitions of DNA states (S3 Fig) [22]. Furthermore, such attenuation of transcriptional noise—while maintaining comparable levels of ultrasensitivity through the addition of repression mechanisms—persists even when the numbers of activators and repressors fluctuate due to processes such as repressor birth–death dynamics and repressor-induced activator degradation (S4 Fig). This effect is robust across a broad parameter range, and stronger repression further enhances noise suppression (S5 Fig). Accordingly, under biologically relevant conditions, only the combination of sequestration, blocking, and displacement exhibits a low Fano factor ranging from 1 to 20—consistent with the experimentally observed range in mammalian cells—and is uniquely capable of reducing it to values close to 1 [37] (Fig 3c).

## Multiple biological oscillators employ a combination of sequestration, blocking, and displacement mechanisms to produce rhythms that are both robust and precise

The combination of sequestration, blocking, and displacement constitutes a core regulatory logic shared across diverse biological oscillators. In the mammalian circadian clock, the PER-CRY complex sequesters CLOCK-BMAL1, displaces it from DNA, and CRY further blocks its transcriptional activity [21,38–44] (Fig 4a). In the NF-κB oscillator, IκB sequesters NF-κB in the cytoplasm, and upon external stimulation, its degradation allows NF-κB to activate transcription; this

**Table 3. The transcriptional activity and the Fano factor for all models describing combinations of indirect repression mechanisms.**

| | | |
|---|---|---|
| The sequestration-based switch | Transcriptional activity | $\dfrac{A(R_T,A_T,K_s)/\Omega K_a}{1+A(R_T,A_T,K_s)/\Omega K_a}$<br><br>where $A\left(R_T,A_T,K_s\right)=\dfrac{A_T-R_T-\Omega K_s+\sqrt{(A_T-R_T-\Omega K_s)^2+4\Omega A_T K_s}}{2}$. |
| | Fano factor | $1+\dfrac{\alpha}{k_f\left(1+A(R_T,A_T,K_s)/\Omega K_a\right)\left(A(R_T,A_T,K_s)/\Omega+K_a+\beta/k_f\right)}$ |
| The sequestration- and blocking-based switch | Transcriptional activity | $\dfrac{\frac{A(R_T,A_T,K_s)}{\Omega K_a}}{1+\frac{A(R_T,A_T,K_s)}{\Omega K_a}+\frac{A(R_T,A_T,K_s)}{\Omega K_a}\frac{R(R_T,A_T,K_s)}{\Omega K_b}}$<br><br>where $R\left(R_T,A_T,K_s\right)=\dfrac{R_T-A_T-\Omega K_s+\sqrt{(A_T-R_T-\Omega K_s)^2+4\Omega A_T K_s}}{2}$. |
| | Fano factor | $1+\dfrac{\alpha\left[K_b+\frac{\beta}{k_f}+\left(\frac{A(R_T,A_T,K_s)}{\Omega}+\frac{\beta}{k_f}\right)\frac{A(R_T,A_T,K_s)}{\Omega K_a}\frac{R(R_T,A_T,K_s)}{\Omega K_b}\right]}{k_f\left(1+\frac{A(R_T,A_T,K_s)}{\Omega K_a}+\frac{A(R_T,A_T,K_s)}{\Omega K_a}\frac{R(R_T,A_T,K_s)}{\Omega K_b}\right)\left[K_a\left(K_b+\frac{\beta}{k_f}\right)+\left(\frac{A(R_T,A_T,K_s)}{\Omega}+\frac{\beta}{k_f}\right)\left(\frac{R(R_T,A_T,K_s)}{\Omega}+K_b+\frac{\beta}{k_f}\right)\right]}$ |
| The sequestration-, blocking- and displacement-based switch | Transcriptional activity | $\dfrac{I(R_T,A_T,K_a,K_s)\frac{A(R_T,A_T,K_s)}{\Omega K_a}}{1+I(R_T,A_T,K_a,K_s)\frac{A(R_T,A_T,K_s)}{\Omega K_a}+J(R_T,A_T,K_a,K_s)\frac{A(R_T,A_T,K_s)}{\Omega K_a}\frac{R(R_T,A_T,K_s)}{\Omega K_b}}$<br><br>where $I\left(R_T,A_T,K_a,K_s\right)=\dfrac{K_s+\sigma K_a+R(R_T,A_T,K_s)/\Omega}{K_s+\sigma K_a+\sigma R(R_T,A_T,K_s)/\Omega}$,<br><br>$J\left(R_T,A_T,K_a,K_s\right)=\dfrac{K_s+K_a+R(R_T,A_T,K_s)/\Omega}{K_s+\sigma K_a+\sigma R(R_T,A_T,K_s)/\Omega}$, and $\sigma=\dfrac{K_s K_d}{K_a K_b}$. |
| | Fano factor | $1+\dfrac{\alpha\left[\left(\frac{R_A(R_T,A_T,K_s)}{\Omega}+K_b+K_d+\frac{\beta}{k_f}\right)+\left(\frac{A_T}{\Omega}+K_d+\frac{\beta}{k_f}\right)J\left(R_T,A_T,K_a,K_s\right)\frac{A(R_T,A_T,K_s)}{\Omega K_a}\frac{R(R_T,A_T,K_s)}{\Omega K_b}\right]}{k_f\left(1+I(R_T,A_T,K_a,K_s)\frac{A(R_T,A_T,K_s)}{\Omega K_a}+J(R_T,A_T,K_a,K_s)\frac{A(R_T,A_T,K_s)}{\Omega K_a}\frac{R(R_T,A_T,K_s)}{\Omega K_b}\right)\left[\left(K_b+K_d+\frac{\beta}{k_f}\right)\left(\frac{A(R_T,A_T,K_s)}{\Omega}+K_d+\frac{\beta}{k_f}\right)+\left(K_b+K_a+\frac{\beta}{k_f}\right)\frac{R_A(R_T,A_T,K_s)}{\Omega}+\left(\frac{A_T}{\Omega}+K_d+\frac{\beta}{k_f}\right)\frac{R(R_T,A_T,K_s)}{\Omega}\right]}$,<br><br>where $R_A\left(R_T,A_T,K_s\right)=\dfrac{A_T+R_T+\Omega K_s-\sqrt{(A_T-R_T-\Omega K_s)^2+4\Omega A_T K_s}}{2}$. |

activation is subsequently suppressed as IκB displaces NF-κB from the DNA [29,45] (Fig 4b). In the p53-Mdm2 oscillator, Mdm2 sequesters p53 away from DNA, blocks transcriptional activity of p53, and displaces it from DNA through cooperative interactions with other corepressors [46,47] (Fig 4c). Although these systems differ in their biological roles—circadian rhythms must sustain precise oscillations under constant conditions, whereas NF-κB and p53 oscillations are transient and stimulus-induced—they employ the same regulatory strategy to achieve functional robustness. The coordinated action of sequestration, blocking, and displacement sharpens the input–output response, creating ultrasensitivity while buffering against fluctuations in upstream signals. We hypothesized that this noise-resistant ultrasensitive switch (Fig 4d, left) enables strong oscillatory responses and precise timing of transitions between transcriptional on and off states (Fig 4d, right) [1,3,4,14], thereby ensuring coherent and stable oscillations once they are triggered.

To validate this hypothesis, we constructed a transcriptional negative feedback loop (NFL) model (Fig 4e and Table 1). In the NFL model, transcriptional activation leads to the synthesis of mRNA ($M$), which is subsequently translated into the repressor in the cytoplasm ($R_c$). Upon nuclear entry, the repressor ($R$) suppresses its own transcription via multiple

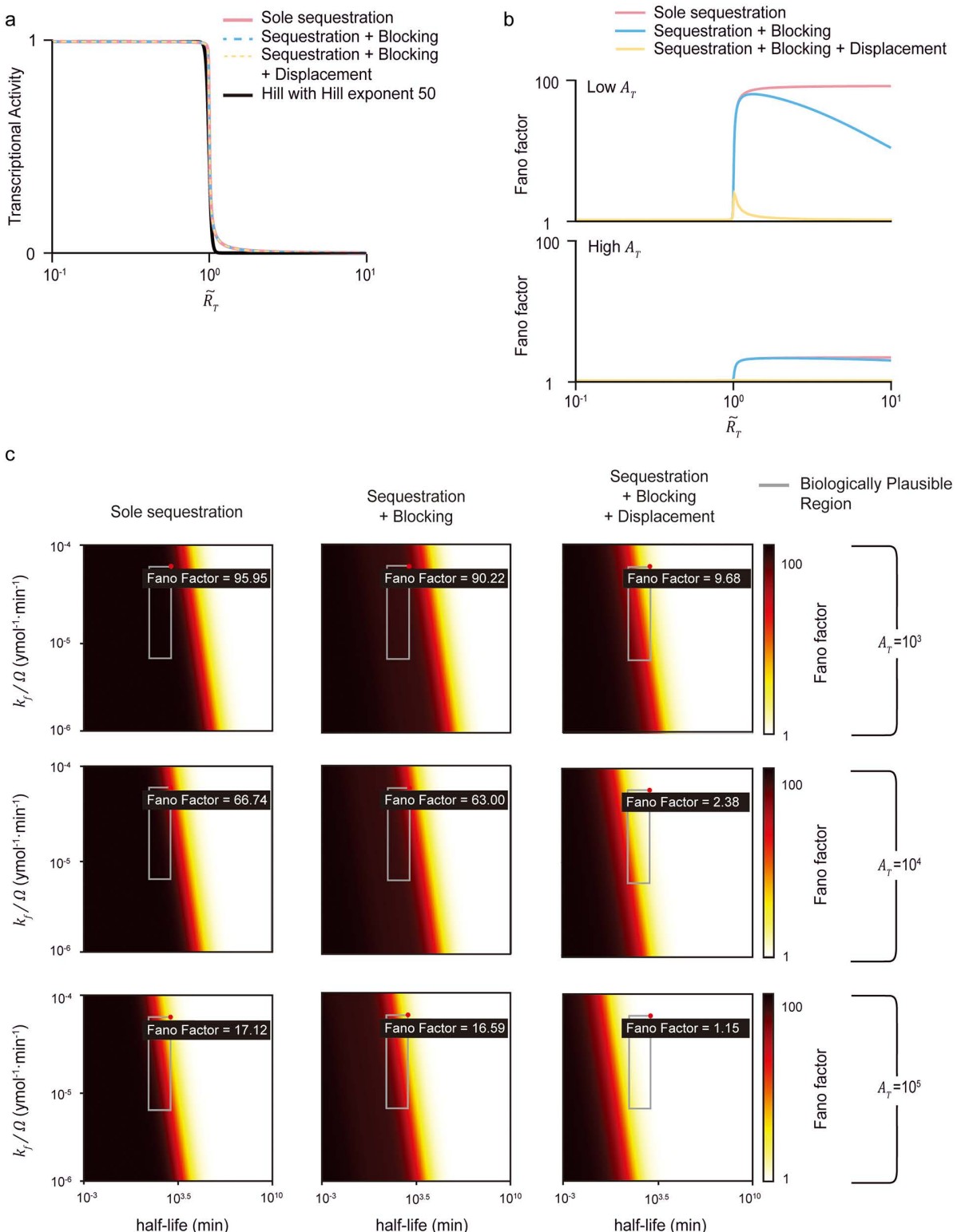

**Fig 3. Only the combination of sequestration, blocking and displacement can achieve close to the minimum noise levels with considerably high sensitivity under biologically realistic conditions.** (a) The transcriptional activities of the sole sequestration (red solid line), the combination of sequestration and blocking (blue dashed line), and the combination of sequestration, blocking, and displacement (yellow dashed line) are matched to

resemble the Hill function with Hill exponent of 50 (black solid line) by adjusting the dissociation constants $K_a$, $K_s$, $K_b$, and $\mathbf{K}_d$. (b) Despite similarities in transcriptional activities, the three models show significant differences in overall noise levels (top). These noise levels decrease with the addition of repression mechanisms, from the sole sequestration (red solid line) to the combination of sequestration and blocking (blue solid line), and further to the combination of sequestration, blocking, and displacement (yellow solid line). Notably, more sensitive responses with Hill exponent of 50 induce greater noise levels compared to those observed with Hill exponent of 4 (Fig 2c, top). Furthermore, increasing $A_T$ leads to an overall reduction in noise levels (bottom). (c) The maximum Fano factor for each model under varying $A_T$ values is shown. When $A_T = 10^3$ (first row), all models show high noise levels (i.e., Fano factor > 9) over the biological range of $k_f\Omega^{-1}$ and the mRNA half-life. Noise levels decrease as $A_T$ increases (2nd and 3rd rows). In particular, when $A_T = 10^5$, the combination of all three mechanisms can lead to reduced noise levels (i.e., a Fano factor close to 1). On the other hand, sole sequestration still results in considerable noise levels (i.e., Fano factor = 17).

repression mechanisms (Fig 4e, gray box). Using this model, we simulated a time-series of mRNA copy numbers ($M(t)$) to investigate the robustness of each repression mechanism against noise (Fig 4f, top). Specifically, we calculated auto-correlation functions of 300 simulated $M(t)$ ($C(t)$; Fig 4f, bottom, green solid line), and fitted them to a decaying cosine function, $\widetilde{C}(t) = e^{-t/\tau} \cdot \cos\frac{2\pi t}{T}$, to estimate $T$ and $\tau$ (Fig 4f, bottom, gray dashed line). By definition of $\widetilde{C}(t)$, $T$ represents the period of the rhythm, while $\tau$ reflects how slowly the autocorrelation decays [48,49]. We observed that, as additional repression mechanisms were incorporated, the variance in the estimated $T$ decreased. It implies more consistent periods in each cycle of rhythm under noise (Fig 4g), indicating enhanced robustness to noise.

## The combination of indirect repressions preserves noise robustness even under multi-target gene regulation

We analyzed models in which 1,000–100,000 transcriptional activators regulate a single target gene (Fig 2a). However, in many biological systems employing multiple repression mechanisms—such as the circadian clock, NF-κB oscillator, and p53-Mdm2 oscillator (Fig 4a–4c)—transcriptional activators often regulate hundreds to thousands of genes simultaneously (Fig 5a). For example, in the mammalian circadian clock, a limited pool of BMAL1 proteins regulates approximately 3,400 genes [50]; p53 controls around 3,700 target genes [51]; and NF-κB targets several hundred genes [52].

Previous studies have suggested that such shared regulation can increase noise, as multiple target genes compete for a limited pool of transcriptional activators [53,54]. To examine whether our proposed mechanisms can overcome this effect, we extended each indirect repression model to include varying numbers of target genes, with regulation mediated through the corresponding repression mechanisms (Fig 5a and Table 1). Notably, increasing the number of target genes has a similar effect as increasing the number of repressors, as it reduces transcriptional activity for the same number of repressors. Thus, for fair comparisons, we evaluated the noise level of each model at equivalent levels of transcriptional activity rather than with equal numbers of repressors (Fig 5b–5d).

Surprisingly, we found that the combination of three indirect repressions preserves noise robustness even under multi-target gene regulation. When the number of the target genes was smaller than that of activators (i.e., 1,000), the transcriptional noise of individual genes remained largely unaffected (Fig 5b–5d, Target gene 0, 10, and 100). Specifically, regardless of repression via the sole sequestration (Fig 5b), the combination of sequestration and blocking (Fig 5c), or the combination of sequestration, blocking, and displacement (Fig 5d), the Fano factor for multiple targets closely matched those of single-target regulation (gray dashed lines). In contrast, when the number of target genes exceeded that of acti-vators (Fig 5b–5d, Target gene 1,000, and 2,000), the Fano factor slightly deviated from the single-target case in the full combination model (Fig 5d), but not in the others (Fig 5b and 5c). This deviation arose because, in the absence of repres-sors, multiple target genes indirectly suppressed each other by depleting the available activator pool (Fig 5d, red-circled points). This competition effectively acts as an additional layer of sequestration, even in the combined repression model, increasing the noise to a level comparable to that of the sole sequestration model. However, as the number of repressors increased (i.e., as the transcriptional activity decreased), the multiple repression effects became dominant. Consequently, the Fano factor converged to that of the single-target case with multiple repressions (Fig 5d). These results indicate that, contrary to the conventional expectation that multi-target regulation amplifies transcriptional noise, the combined

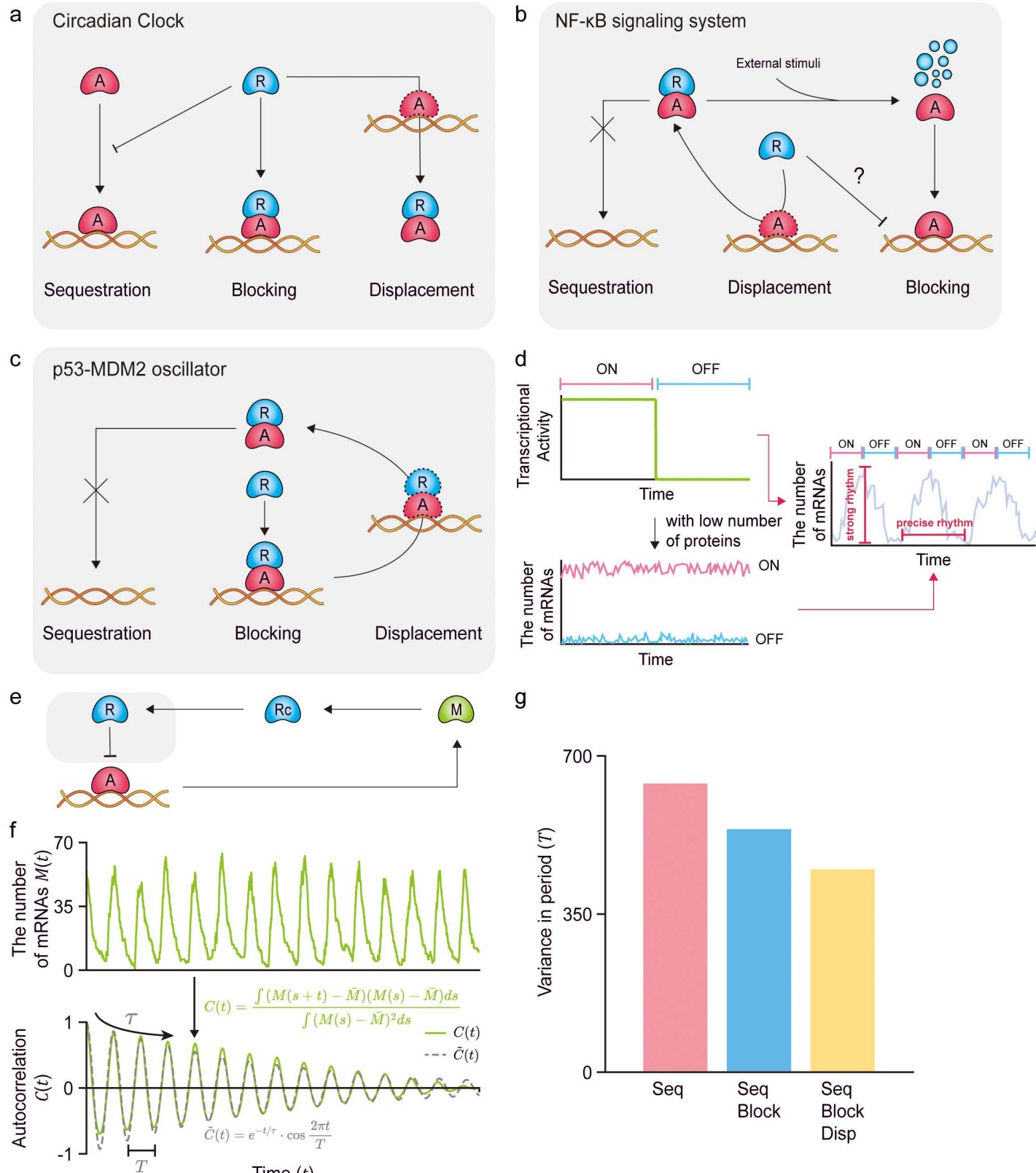

**Fig 4. Various biological oscillators utilize the combination of sequestration, blocking, and displacement mechanisms to generate robust and precise rhythms.** (a) In the mammalian circadian clock, PER:CRY inhibits CLOCK:BMAL1 by sequestration, preventing its binding to DNA. On the

other hand, when CLOCK:BMAL1 is already bound to DNA, CRY blocks transcription or PER:CRY can displace it from DNA. (b) In the NF-κB signaling system, IκB sequesters NF-κB in the cytoplasm, preventing its entry into the nucleus. As IκB degrades in response to external stimuli, the previously sequestered NF-κB is released and enters the nucleus. In the nucleus, NF-κB binds to DNA to promote transcription. IκB can inhibit transcription by displacing NF-κB from DNA, although whether IκB can directly block transcription remains unclear. (c) In the p53-MDM2 oscillator, MDM2 binds to p53, preventing its binding to DNA. On the other hand, when p53 is already bound to DNA, MDM2 inhibits transcription by directly binding to the DNA-bound p53 to block it, as well as displacing it from DNA in cooperation with a corepressor. (d) The combination of sequestration, blocking, and displacement can generate both high sensitivity and robustness against noise, essential for strong and precise rhythms, respectively. Furthermore, even with a low number of proteins (i.e., activators and repressors), mRNAs exhibit low fluctuations in both on and off transcriptional states. Consequently, biological oscillators incorporating all three mechanisms can effectively produce precise rhythms, with accurate increases and decreases during the transcriptional on and off states, respectively. (e) To investigate this, we constructed the transcriptional negative feedback loop (NFL) model. In the NFL model, when the transcription is turned on, mRNA ($M$) is produced, and subsequently translated into the cytoplasmic repressor ($R_c$). After entering the nucleus, the nucleic repressor ($R$) inhibits its own transcription through the multiple repression mechanisms (gray box). (f) Using the NFL model, the oscillatory time-series of mRNA copy numbers ($M(t)$) can be simulated (top). To quantify the noise level in $M(t)$, its autocorrelation function $C(t)$ (bottom, green solid line) is computed and is fitted with a decaying cosine function $\tilde{C}(t) = e^{-t/\tau} \cdot \cos\frac{2\pi t}{T}$ (bottom, gray dashed line). Here, $T$ represents the period of oscillation, and $\tau$ represents how slow $C(t)$ decays. (g) By simulating $M(t)$ for three models over 300 times, we evaluated the variance of $T$. As more repression mechanisms are combined, the variance of $T$ is reduced, indicating enhanced robustness against noise.

sequestration, blocking, and displacement mechanisms can buffer against such fluctuations, enabling a finite pool of activators to robustly regulate large gene networks.

## Discussion

Previous work demonstrated that combining indirect repression mechanisms—such as sequestration, blocking, and displacement—can generate ultrasensitive transcriptional switches that are more robust to molecular noise than direct mechanisms such as cooperative binding [22]. However, those studies did not account for the timescales of DNA-binding dynamics (i.e., the binding and unbinding rates between DNA and transcriptional factors). Here, we show that this omission can be misleading: when DNA binding occurs on a faster timescale than mRNA-related processes (i.e., transcription and degradation), ultrasensitivity and noise robustness can be simultaneously achieved in the resulting transcriptional switch (Fig 1b–1e, 2b, and d), regardless of the underlying repression mechanism. We further show that increasing the number of activators can also promote such ultrasensitive behavior (Fig 2b–2d). However, under realistic constraints on binding kinetics and activator abundance, robust ultrasensitive switching is only achieved when all three indirect repression mechanisms are combined (Fig 3). As a result, biological oscillators incorporating this triple-repression architecture can maintain precise rhythmic transitions between transcriptional states even in the presence of molecular noise (Fig 4). In summary, our findings suggest that combining indirect repression mechanisms offers a resource-efficient strategy for achieving robust, rhythmic gene expression in noisy cellular environments—potentially explaining why sequestration, blocking, and displacement often co-occur in natural biological oscillators.

This strategy is exemplified in systems such as NF-κB oscillators, which often exhibit heterogeneous single-cell oscillatory responses [55]. This heterogeneity primarily results from differences in upstream stimulus strength or duration, such as variations in inflammatory stimuli TNF-α pulse strength or duration that modulate the timing of NF-κB activation [56]. Similarly, in the p53-MDM2 oscillators, cell-to-cell variability in oscillatory dynamics is induced by differences in DNA damage or oncogenic stress [57]. These demonstrate that variability in input stimuli primarily determines when and how cells cross the activation threshold, required to initiate oscillations [58,59]. Therefore, our findings are primarily relevant to the regime where oscillations are triggered by sufficiently strong inputs, and to how the identified repression mechanisms help maintain coherent and stable rhythms of initiated oscillations. Extending this framework to variable input conditions, where upstream stimuli couple with repression strategies thus presents a compelling future research direction for understanding population-level transcriptional dynamics.

Although multiple indirect repression mechanisms may help improve population-level coherence and enhance noise resilience, these advantages could come at a cost at the single-cell level, particularly when compared with direct

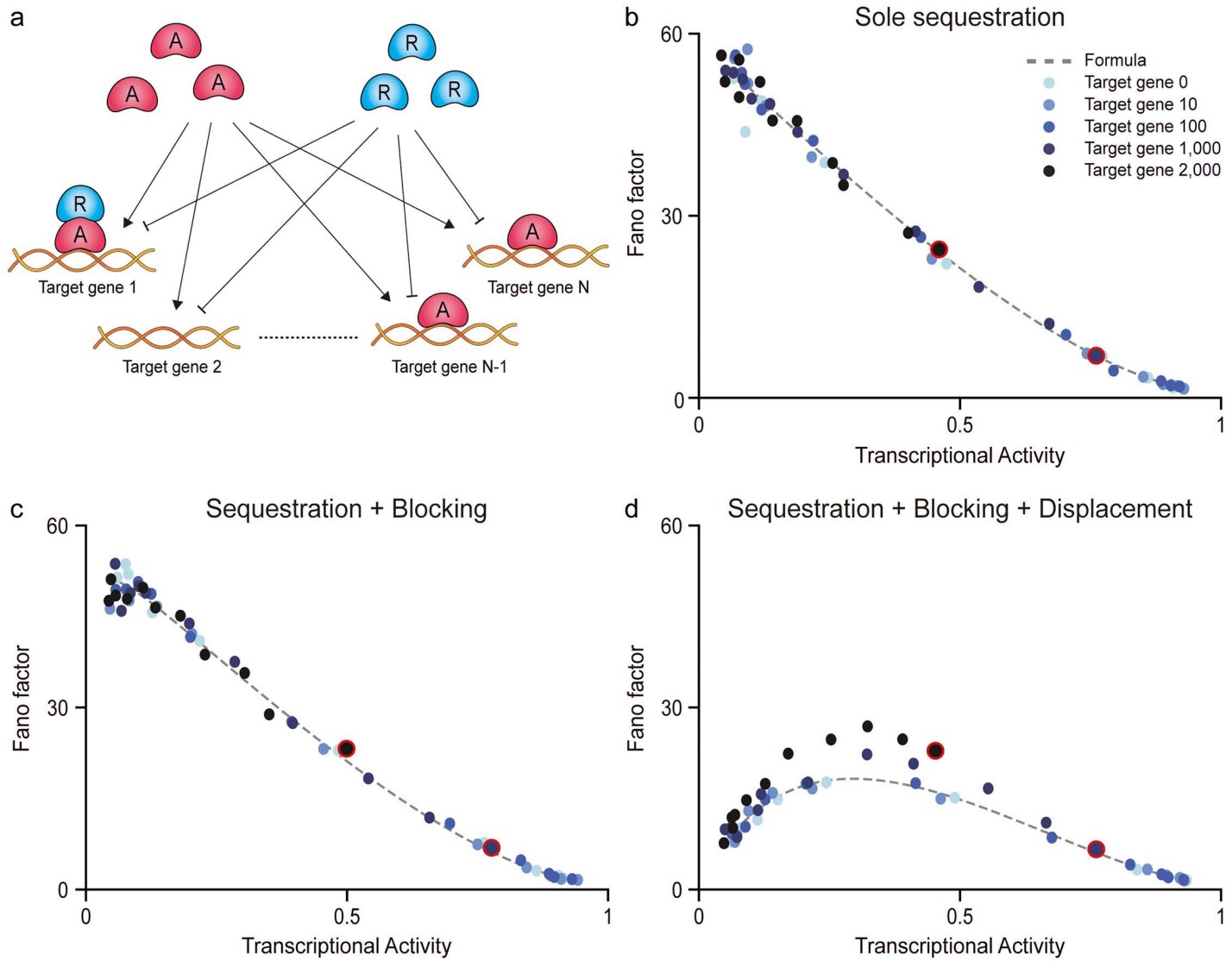

**Fig 5. Transcriptional factors regulate thousands of target genes while maintaining noise levels comparable to those of single-gene regulation, given the same level of transcriptional activity.** (a) In biological systems, transcriptional activators and repressors often regulate multiple target genes simultaneously. (b–d) We examine whether combining different repression mechanisms can still reduce transcriptional noise. For this, we simulated the stationary distribution of mRNA of one target gene while varying the number of additional target genes to 0, 10, 100, 1,000, and 2,000 (see Methods). From the distribution, we calculated the Fano factor of mRNAs (blue dots). Regardless of the number of target genes, repression via the sole sequestration (b), the combination of sequestration, and blocking (c), the combination of sequestration, blocking, and displacement (d), all show transcriptional noise levels similar to single-gene regulation. Here, dashed lines represent the relationship between the transcriptional activity and the Fano factor without additional target genes.

repressor–DNA binding. Specifically, indirect repression mechanisms inherently require the production of additional proteins (i.e., activators), thereby imposing higher energetic and biosynthetic demands on the cell. The synthesis and maintenance of these additional proteins consume substantial biosynthetic resources, and they can limit the cell's capability to produce other essential proteins, which the cell requires to function properly [60–62]. As a result, it is plausible that natural selection often favors simpler regulatory strategies such as direct repressor-DNA binding, which provide efficient

transcriptional control with lower biosynthetic cost, even if this comes at the cost of reduced noise resilience and ultrasensitivity. This may explain why many biological systems have evolved to select a direct repressor-DNA binding mechanism [63]. Overall, this trade-off between functional performance and resource limitation may represent a fundamental design principle in transcriptional regulation, and investigating how cells balance these competing demands offers a promising avenue for future research.

Indeed, recent studies have shown that consuming resources, i.e., energy, can enhance functional performance, including ultrasensitivity and noise-resilience, in biological oscillators. For example, we show that, without energy consumption (i.e., under the detailed balance condition $\frac{K_a K_b}{K_s K_d} = 1$) [21], simply combining indirect repression mechanisms can enhance the noise robustness of biological oscillators (Fig 4). However, recent studies suggest that breaking detailed balance through energy dissipation can further amplify ultrasensitivity. For example, Jeong et al. demonstrated that violating detailed balance in triple-mechanism models yields greater ultrasensitivity than in other combinations [21]. Similarly, Estrada et al. showed that in cooperative binding systems, breaking detailed balance facilitates ultrasensitive responses [64]. Moreover, energy consumption has also been linked to enhanced noise robustness in oscillatory systems: Cao et al. theoretically showed that energy dissipation suppresses phase diffusion due to intrinsic noise in various biochemical oscillators [49], while Fei et al., using the Brusselator model, found that energy expenditure not only reduces phase diffusion but also improves adaptability to environmental changes [65]. Collectively, these findings highlight that beyond combining repression mechanisms, the energy cost of regulatory processes plays a key role in tuning ultrasensitivity and robustness. Investigating how energy dissipation interacts with transcriptional repression strategies thus presents another compelling direction for future research.

Beyond transcriptional repression, several non-transcriptional mechanisms have also been shown to generate ultrasensitive responses in biological systems. For instance, multisite binding without cooperativity can produce ultrasensitivity when coupled with other processes such as differential degradation or modification rates among molecular complexes [66,67]. Similarly, post-transcriptional regulation through microRNA-mediated repression or RNA-binding proteins can give rise to threshold-like responses and tune gene expression sensitivity [68,69]. At the post-translational level, protein modification cycles—such as phosphorylation–dephosphorylation cascades or ubiquitination [15,16,19]—can further amplify response sensitivity and fine-tune the dynamic range through mechanisms such as enzyme saturation [2,70]. Therefore, investigating how these non-transcriptional mechanisms operating at translational, post-transcriptional, and post-translational levels influence transcriptional noise would be an interesting direction for future research.

Finally, we derived analytical expressions for the mean (i.e., transcriptional activity) and the Fano factor of mRNA across different repression models. While our analysis focused on these stationary statistics, recent analytical advances suggest that more detailed stochastic quantities can also be obtained. Specifically, by applying multiscale simplification techniques [71], the full probability distribution of mRNA for the multi–binding-site model (Fig 1a) could in principle be derived. Similarly, the autocorrelation function of the oscillator model, which we used to characterize noisy rhythmic properties in biological oscillators (Fig 4), could be derived using linear noise approximation [72]. Extending our framework in these directions would deepen understanding of the stochastic underpinnings of gene regulation and provide additional validation for the theoretical conclusions drawn here.

## Methods

### Derivation of the equations for the transcriptional activity and the Fano factor of the four binding sites model with cooperative binding

To derive the transcriptional activity and Fano factor of a gene regulated by cooperative binding at four DNA sites, we followed the framework introduced by Sanchez et al. [28], formulating a chemical master equation (CME) that captures all 16 possible DNA binding configurations. By computing steady-state moment equations for mRNA distributions, we obtained closed-form expressions for the transcriptional activity and Fano factor as functions of repressor concentration

and binding parameters. These derivations quantify how cooperative repression influences gene expression dynamics, and the full mathematical details are provided in the Supplementary Information and at https://github.com/Mathbiomed/Ultrasensitive-gene-switch.

**Derivation of the equations for the transcriptional activity and the Fano factor of the multiple indirect repression mechanism models**

Following the approach in the previous section, Jeong et al., derived equations for the transcriptional activity and the Fano factor in models incorporating the multiple indirect repression models [22]. Notably, for analytical convenience, all parameters were normalized by $(k_f \Omega^{-1}) A_T$. For example, under this normalization, the transcription and degradation rates of mRNA, $\alpha$ and $\beta$, were simply switched to $\widetilde{\alpha} = \alpha\Omega/k_f A_T$ and $\widetilde{\beta} = \beta\Omega/k_f A_T$, respectively, and the unbinding rate between activators and DNA, $k_a$, was expressed as the normalized dissociation constant, $\widetilde{K_a} = \Omega K_a/A_T$. In particular, the binding of activators $(k_f \Omega^{-1}) A(R_T, A_T, K_s)$, where the number of free activators is given by

$$A(R_T, A_T, K_s) = \frac{A_T - R_T - \Omega K_s + \sqrt{(A_T - R_T - \Omega K_s)^2 + 4\Omega A_T K_s}}{2},$$

was normalized to the fraction of free activators among the total activators,

$$\widetilde{A}\left(\widetilde{R_T}, \widetilde{K_s}\right) = \frac{A(R_T, A_T, K_s)}{A_T} = \frac{1 - \widetilde{R_T} - \widetilde{K_s} + \sqrt{\left(1 - \widetilde{R_T} - \widetilde{K_s}\right)^2 + 4\widetilde{K_s}}}{2}$$

where $\widetilde{R_T} = R_T/A_T$ and $\widetilde{K_s} = \Omega K_s/A_T$ [14,22,35,73–76]. With the normalized variables and parameters, the transcriptional activity and the Fano factor for the sole sequestration model were derived as follows:

$$TA\left(\widetilde{R_T}\right) = \frac{\widetilde{A}\left(\widetilde{R_T}, \widetilde{K_s}\right)/\widetilde{K_a}}{1 + \widetilde{A}\left(\widetilde{R_T}, \widetilde{K_s}\right)/\widetilde{K_a}},$$

$$FF\left(\widetilde{R_T}\right) = 1 + \frac{\widetilde{\alpha}}{\left(1 + \widetilde{A}\left(\widetilde{R_T}, \widetilde{K_s}\right)/\widetilde{K_a}\right)\left(\widetilde{A}\left(\widetilde{R_T}, \widetilde{K_s}\right) + \widetilde{K_a} + \widetilde{\beta}\right)}.$$

To incorporate $k_f$, $A_T$, and $\Omega$ to the equations explicitly, we substitute $\widetilde{\alpha}$, $\widetilde{\beta}$, $\widetilde{R_T}$, $\widetilde{K_a}$, and $\widetilde{K_s}$ with the original parameters as follows:

$$TA(R_T) = \frac{A(R_T, A_T, K_s)/\Omega K_a}{1 + A(R_T, A_T, K_s)/\Omega K_a},$$

$$FF(R_T) = 1 + \frac{\alpha}{k_f(1 + A(R_T, A_T, K_s)/\Omega K_a)(A(R_T, A_T, K_s)/\Omega + K_a + \beta/k_f)}.$$

Similarly, we substituted the original parameters into the transcriptional activity and the Fano factor equations derived in Jeong et al for other combinations of indirect repression mechanisms (Table 3).

## Simulations of multi-target genes regulation and noise level quantification

To evaluate how transcriptional noise is affected when transcriptional activators regulate multiple target genes, we extended the multiple indirect repression models to include up to 2,000 co-regulated genes (Fig 5). To isolate the effect of multi-gene regulation, we fixed the total number of activators to 1,000, while the number of additional target genes was varied from 0 to 2,000. For each gene count, we modulated the transcriptional activity by varying the number of repressors from 0 to 2,000. For each parameter set, we simulated the corresponding CME (Table 1) using the Gillespie algorithm with 1,000 independent runs. The transcriptional activity (i.e., the probability that the gene is active) was quantified as the proportion of runs in which $E_A = 1$ at the stationarity (Fig 5b–5d, x-axis). Similarly, from resulting stationary mRNA distributions, we calculated the Fano factor to measure the transcriptional noise (Fig 5b–5d, y-axis).

## Supporting information

**S1 Text. Derivation of the equations for the transcriptional activity and the Fano factor of the four binding sites model with cooperative binding.**
(DOCX)

**S1 Fig. The cooperative binding mechanism, in which activator binding activates transcription, is sensitive to noise.** (a) Model diagram of the transcription regulated by the activator proteins ($A$) binding to four independent sites on the DNA within a cell volume of $\Omega$. Each binding site is occupied by $A$ at a rate $k_f$. Conversely, $A$ unbinds from DNA at a rate $k_a$ when one site is occupied, with the dissociation constant between $A$ and DNA defined as $K_a = \frac{k_a}{k_f}$. For two, three, or four occupied sites, $A$ unbinds at rates $ck_a$, $c^2k_a$ and $c^3k_a$, respectively. Accordingly, when $c < 1$, cooperative binding is present. When all binding sites are occupied, mRNA is produced at a rate $\alpha$ and degrades at a rate $\beta$, whereas transcription is inactive if any site remains unoccupied. (b) When $c = 10^{-2}$, transcriptional activity closely resembles the Hill function with a Hill exponent of 4 (black line). Furthermore, transcriptional activity remains consistent with the Hill function regardless of $k_f$ values, provided $K_a$ is kept constant (red dotted line and blue dashed line). (c) Nevertheless, the overall noise level quantified by the Fano factor of mRNAs shows significant differences based on $k_f$. Notably, the overall noise levels are reduced during the sensitive response when $k_f$ is faster (red solid line) compared to when it is slower (blue solid line).
(TIF)

**S2 Fig. The original parameter set does not fully satisfy the conditions required for the quasi–steady-state approximation (QSSA).** (a) Ten representative time-series of mRNA copy numbers (thin lines) and their mean trajectories (thick lines) were obtained from 1,000 stochastic simulations using the model that combines sequestration, blocking, and displacement. The reduced model, in which the numbers of activators and repressors are approximated by their QSSA (gray lines; see Methods), was compared with the full model that explicitly models the binding and unbinding reactions between them (green lines). Simulations were performed while varying the molar ratio $\widetilde{R_T} = 0.2$ (left), $1$ (middle), and $5$ (right), with the original parameter set in Table 1. (b) Probability density functions of the simulated mRNA copy numbers at $500$ h, obtained from 1,000 stochastic simulation runs varying the molar ratio $\widetilde{R_T} = 0.2$ (left), $1$ (middle), and $5$ (right). The full and reduced models show consistent dynamics at each $\widetilde{R_T}$.
(TIF)

**S3 Fig. Simulated time-series of DNA states and mRNA copy numbers, and stationary distribution of mRNA copy numbers across the three repression models.** (a–c) Stationary distributions of mRNA copy numbers for the sole sequestration (a), combined sequestration and blocking (b), and combined sequestration, blocking, and displacement models (c), simulated using the parameters in Fig 3 at $\widetilde{R_T} = 5$. Despite the large number of repressors compared to the activators, both the sole sequestration model and the combined sequestration and blocking model exhibited bimodal

mRNA distributions, with peaks at both low and high mRNA numbers, indicating that the activated DNA state persisted even in the presence of many repressors. In contrast, incorporation of displacement rendered the mRNA distribution unimodal, reflecting more consistent transcriptional repression. (d-i) Even under a large excess of repressors over activators, the sequestered activator can stochastically dissociate and rebind to DNA to form the active complex $E_A$ (d–f(i), at time 0). In the sole sequestration model (d), once the activator binds to DNA to form $E_A$ (d(i)), it remains bound for a long duration before dissociating into $E_F$ (d(ii)). This leads to continuous mRNA accumulation (g), and thus a bimodal mRNA distribution with a high Fano factor. In the combined sequestration and blocking model (e), $E_A$ rapidly transitions to the repressed state $E_R$ via blocking (e(i) and (ii)), and frequently interconverts between $E_A$ and $E_R$ until the activator dissociates from DNA to form $E_F$ (e(iii)). These frequent blocking events delay mRNA accumulation (h), and reduce the separation between the two peaks in the mRNA distribution, resulting in a lower Fano factor than in the sole sequestration model. In contrast, in the combined sequestration, blocking, and displacement model (f), $E_A$ rapidly transitions to $E_R$ (f(i) and (f(ii))) and displacement accelerates the transition from $E_R$ to $E_F$ (f(iii)), thereby preventing mRNA accumulation (i) and producing a unimodal mRNA distribution with a low Fano factor.
(TIF)

**S4 Fig. Despite fluctuations in repressor or activator copy number, the combined sequestration, blocking, and displacement model maintains robustness of noise reduction.** (a) When a repressor is regulated by a simple birth–death process, its copy number ($R_T$) follows a Poisson distribution with the mean of $< R_T >$. (b) In this case, the total mean of the mRNA copy number ($M$) can be calculated through the law of the total mean, $E[M] = E_\pi [E [M|R_T]]$, where denotes the probability mass function of Poisson distribution with the mean of $< R_T >$ and $E[M|R_T]$ is the conditional mean of $M$ for a given $R_T$. As $E_\pi [E[M|R_T]]$ can be calculated as the product of the production-to-degradation rate ratio ($\alpha/\beta$) and the transcriptional activity $TA(R_T)$ in Table 2 (i.e., $E[M|R_T] = \frac{\alpha}{\beta} TA(R_T)$), the total mean of $M$ becomes $E[M] = E_\pi \left[ \frac{\alpha}{\beta} TA(R_T) \right] = \frac{\alpha}{\beta} \int TA(R_T) \pi (R_T; < R_T >) dR_T$. Because the transcriptional activity $TA(R_T)$ was set to be identical across all indirect repression models, their total mean mRNA numbers—and thus their effective transcriptional activities under repressor fluctuation (i.e., the total mean multiplied by $\beta/\alpha$)—are also identical across models: sole sequestration (red), combined sequestration and blocking (blue), and combined sequestration, blocking, and displacement (yellow). (c) Similarly, the total variance of the mRNA copy number can be calculated through the law of the total variance, $Var[M] = Var_\pi [E[M|R_T]] + E_\pi [Var[M|R_T]]$, where $Var[M|R_T]$ is the conditional variance of $M$ for a given $R_T$. As $Var[M|R_T]$ can be calculated as the product of the product of $E[M|R_T]$ and the Fano factor $FF(R_T)$ in Table 2 (i.e., $\frac{\alpha}{\beta} TA(R_T) FF(R_T)$), the total variance of $M$ becomes $Var[M] = \frac{\alpha}{\beta} \int (TA(R_T) - E[M])^2 \pi (R_T; < R_T >) dR_T + \frac{\alpha}{\beta} \int TA(R_T) FF(R_T) \pi (R_T; < R_T >) dR_T$. Because the transcriptional activity $TA(R_T)$ was set to be identical across all indirect repression models, the first term is also the same among models. In contrast, the second term differs due to variations in the Fano factor $FF(R_T)$. Consequently, consistent with its lowest Fano factor $FF(R_T)$, the model combining sequestration, blocking, and displacement exhibits the smallest overall mRNA variance, and thus the lowest overall mRNA Fano factor, demonstrating its robustness even under fluctuations in repressor copy number. (d) The number of activators can also fluctuate, when repressors bind to and promote the degradation of activators, as observed in systems such as the p53–MDM2 oscillator. To incorporate such fluctuations, we modeled the birth-death dynamics of the activator. Specifically, activators are produced at a rate $\alpha_A$, degraded as free activators at a rate $\beta_A A/\Omega$, and further degraded in the form of repressor–activator complexes at a rate $\beta_{RA} R_A/\Omega$. Here, $A(A_T; R_T) = \frac{A_T - R_T - \Omega K_s + \sqrt{(A_T - R_T - \Omega K_s)^2 + 4\Omega K_s A_T}}{2} \approx \begin{cases} 0, & A_T \le R_T \\ A_T - R_T, & A_T > R_T \end{cases}$, and $R_A(A_T; R_T) = \frac{A_T + R_T + \Omega K_s - \sqrt{(A_T - R_T - \Omega K_s)^2 + 4\Omega K_s A_T}}{2} \approx \begin{cases} A_T, & A_T \le R_T \\ R_T, & A_T > R_T \end{cases}$ represent the number of free activators and repressor–activator complexes, respectively, as functions of the activator copy number ($A_T$) for a given $R_T$. Then, the probability distribution of $A_T$ is governed by the following CME. $\frac{dP(A_T)}{dt} = \alpha_A P(A_T - 1) + \left( \beta_A \frac{A(A_T + 1)}{\Omega} + \beta_{RA} \frac{R_A(A_T + 1)}{\Omega} \right) P(A_T + 1) - \left( \alpha_A + \beta_A \frac{A(A_T)}{\Omega} + \beta_{RA} \frac{R_A(A_T)}{\Omega} \right) P(A_T)$, where $P(A_T)$ denotes

the probability that the total number of activators is $A_T$, At stationary distribution (i.e., $\frac{dP(A_T)}{dt} = 0$), the CME satisfies

$$P(A_T + 1) = \frac{\frac{\alpha_A \Omega}{\beta_{RA}}}{\frac{\beta_A}{\beta_{RA}} A(A_T; R_T) + R_A(A_T; R_T)} P(A_T) \approx \begin{cases} \frac{\frac{\alpha_A \Omega}{\beta_{RA}}}{A_T} P(A_T) \\ \frac{\frac{\alpha_A \Omega}{\beta_{RA}}}{\frac{\beta_A}{\beta_{RA}} A_T + \left(1 - \frac{\beta_A}{\beta_{RA}}\right) R_T} P(A_T) \end{cases}$$

. Thus, the stationary distribution of $A_T$ can be

calculated recursively, and is determined by $\alpha_A \Omega / \beta_{RA}$ and $\beta_{RA}/\beta_A$, which represent the effective number of the activator and the relative degradation activator mediated by the repressor, respectively. Here, we simulated with $\alpha_A \Omega / \beta_{RA} = 1,000$ and $\beta_{RA}/\beta_A = 10$, while varying $R_T$ to modulate the effective molar ratio $R_T^{eff} = R_T / \frac{\alpha_A \Omega}{\beta_{RA}}$. Notably, when the degradation rates of activators alone and in the complex are identical (i.e., $\beta_A = \beta_{RA}$), $A$T follows a simple Poisson distribution. In contrast, when the repressor promotes activator degradation by binding (i.e., $\beta_A < \beta_{RA}$), the stationary distribution of $A_T$ deviates from Poisson. (e) With the calculated non-Poisson distribution of $A_T$ ($\pi(A_T; R_T, \frac{\alpha_A \Omega}{\beta_{RA}}, \frac{\beta_{RA}}{\beta_A})$), the total mean of $M$ is given by $E[M] = E_\pi \left[ \frac{\alpha}{\beta} TA(A_T) \right] = \frac{\alpha}{\beta} \int TA(A_T) \pi \left( A_T; R_T, \frac{\alpha_A \Omega}{\beta_{RA}}, \frac{\beta_{RA}}{\beta_A} \right) dA_T$. Because the transcriptional activity $TA(A_T)$ was set to be identical across all indirect repression models, their total mean mRNA numbers—and thus their effective transcriptional activities under activator fluctuation (i.e., the total mean multiplied by $\beta/\alpha$)—are also identical across models: sole sequestration (red), combined sequestration and blocking (blue), and combined sequestration, blocking, and displacement (yellow). (f) Similarly, the total variance of the mRNA copy number can be calculated through the law of the total variance, $Var[M] = \frac{\alpha}{\beta} \int (TA(A_T) - E[M])^2 \pi \left( A_T; R_T, \frac{\alpha_A \Omega}{\beta_{RA}}, \frac{\beta_{RA}}{\beta_A} \right) dR_T + \frac{\alpha}{\beta} \int TA(A_T) FF(A_T) \pi \left( A_T; R_T, \frac{\alpha_A \Omega}{\beta_{RA}}, \frac{\beta_{RA}}{\beta_A} \right) dA_T$. Because the transcriptional activity $TA(A_T)$ was set to be identical across all indirect repression models, the first term is also the same among models. In contrast, the second term differs due to variations in the Fano factor $FF(A_T)$. Consequently, consistent with its lowest Fano factor $FF(A_T)$, the model combining sequestration, blocking, and displacement exhibits the smallest overall mRNA variance, and thus the lowest overall mRNA Fano factor, demonstrating its robustness even under fluctuations in activator copy number.
(TIF)

**S5 Fig. Systematic evaluation of the effects of indirect repression mechanisms on ultrasensitivity of transcriptional activity and overall noise level.** (a - b) Heatmaps showing the ultrasensitivity of transcriptional activity and the overall noise level, quantified by the Hill coefficient (a) and the area under the Fano factor curve (AUC) over a range of $\widetilde{R_T}$ from $10^{-1}$ to $10^1$ (b), respectively, by varying the dissociation constants $K_a$ and $K_s$ in the three indirect repression models. In the sole sequestration model, strong sequestration (i.e., $K_s \ll K_a$) promoted high ultrasensitivity (a(i)) but was accompanied by elevated noise levels (b(i)). Incorporating blocking, with $K_b = 10^{-3}$, maintained a similar level of ultrasensitivity as in the sole sequestration case (a(ii)), while reducing the overall noise (b(ii)), indicating that the addition of blocking dampens fluctuations without compromising sensitivity. Further strengthening the blocking ($K_b = 10^{-4}$) lowered the noise even more (b(iii)) without loss of ultrasensitivity (a(iii)). Regardless of blocking strength, adding displacement, with its rate set to $K_d = K_a K_b / K_s$ to maintain comparable ultrasensitivity (a(iv-v)), led to additional noise reduction (b(iv-v)), demonstrating cumulative noise suppression through cooperative multiple repressions. Moreover, stronger displacement ($K_d = 10 K_a K_b / K_s$) produced higher ultrasensitivity (a(vi-vii)) with sustained low noise (b(vi-vii)) compared to the weaker displacement ($K_d = K_a K_b / K_s$; a(iv-v) and b(iv-v)). Taken together, the sequential addition and strengthening of repression mechanisms progressively reduced noise while retaining or amplifying ultrasensitivity.
(TIF)

## Author contributions

**Conceptualization:** Eui Min Jeong, Chang Yoon Chung, Jae Kyoung Kim.

**Data curation:** Eui Min Jeong, Chang Yoon Chung.

**Formal analysis:** Eui Min Jeong, Chang Yoon Chung, Jae Kyoung Kim.

**Funding acquisition:** Eui Min Jeong, Jae Kyoung Kim.

**Investigation:** Eui Min Jeong, Chang Yoon Chung, Jae Kyoung Kim.

**Methodology:** Eui Min Jeong, Chang Yoon Chung, Jae Kyoung Kim.

**Project administration:** Jae Kyoung Kim.

**Resources:** Eui Min Jeong, Jae Kyoung Kim.

**Software:** Eui Min Jeong, Chang Yoon Chung.

**Supervision:** Jae Kyoung Kim.

**Validation:** Eui Min Jeong, Jae Kyoung Kim.

**Visualization:** Eui Min Jeong, Chang Yoon Chung.

**Writing – original draft:** Eui Min Jeong, Chang Yoon Chung, Jae Kyoung Kim.

**Writing – review & editing:** Eui Min Jeong, Chang Yoon Chung, Jae Kyoung Kim.

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
