## [Decision Letter · Decision Letter 0]

12 Aug 2025

PCOMPBIOL-D-25-01132

How Cells Tame Noise While Maintaining Ultrasensitive Transcriptional Responses

PLOS Computational Biology

Dear Dr. Kim,

Thank you for submitting your manuscript to PLOS Computational Biology. After careful consideration, we feel that it has merit but does not fully meet PLOS Computational Biology's publication criteria as it currently stands. Therefore, we invite you to submit a revised version of the manuscript that addresses the points raised during the review process.

Please submit your revised manuscript within 60 days Oct 12 2025 11:59PM. If you will need more time than this to complete your revisions, please reply to this message or contact the journal office at ploscompbiol@plos.org. Please include the following items when submitting your revised manuscript:

We look forward to receiving your revised manuscript.

Kind regards,

Christopher E Miles

Academic Editor

PLOS Computational Biology

Mark Alber

Section Editor

PLOS Computational Biology

**Journal Requirements:**

**Reviewers' comments:**

Reviewer's Responses to Questions

Reviewer #1: In the manuscript ‘How Cells Tame Noise While Maintaining Ultrasensitive Transcriptional Responses’, Jeong et al. performed mathematical modeling of transcriptional regulations with several types of mechanisms, including direct DNA binding, sequestration, blocking and displacement. They found that under biologically plausible parameter settings, the combination of three indirect repression mechanisms, but not other options, can maintain low noise of mRNA product while achieving high ultrasensitivity of the responses. They further showed that biological oscillators use this combination to achieve precise control of rhythmic dynamics in the precise of noise and large numbers of target genes. Overall, this is a rigorous study with very interesting observations. I have one suggestion for deepening the insight into the combination of the three mechanisms and a few others for improving the clarity of the manuscript.

1. The study has limited information on how the 3 indirect repression mechanisms synergize to achieve noise reduction and ultrasensitivity maintenance. While I understand this is a challenging question, readers would expect that the authors provide some insight since this is the main conclusion and the authors’ models have provided a foundation for a mechanistic investigation.

1a) The authors can first look more carefully at the distributions of each species in Fig 2a with critical total repressor concentrations and deduce the activities of the 3 mechanisms.

1b) With 1a and other analysis, the authors can clarify whether the Blocking component is essential for the performance by itself (e.g. there is a crucial amount of A-R-DNA complex) or merely serves as a mediator for displacement (e.g. a good performance is achieved when A is always displaced from DNA when B binds).

1c) By tuning the rate constants for the 3 mechanisms continuously and evaluating the performance of the model, the authors can investigate the effects of the 3 mechanisms at a deeper level.

2. The scope of the work and the proper introduction of the system with biological context need to be better presented at the beginning. For example, ‘One of the most well-known mechanisms that produce ultrasensitive responses is cooperative binding, in which transcriptional repressors bind cooperatively to multiple sites on DNA.’ appears as the second sentence of the manuscript, but cooperativity and ultrasensitive responses do not even have to involve transcription in biology: many forms of binding involving small molecules, proteins and RNAs can produce those phenomena.

3. Line 108: ‘To investigate how much faster _f and _r attenuate transcriptional noise…’. This part of the sentence is confusing. Do these parameters accelerate noise reduction, or are the authors investigating the range of them that would reduce noise?

4. Each modeling framework (ODE? CME? Stochastic simulations?) needs to be concisely introduced before the first result plot (e.g. Fig 1b) in addition to the wiring diagram.

5. Many plots in this work have overlapped curves. The visualization of such data needs to be improved otherwise it is difficult for readers to distinguish different models. I suggest that the authors use different line widths for different models and use darker and thinner curves for those in front.

6. Please clarify in the text what type of cell the cell volume parameter corresponds to. Similarly, the ranges of mRNA half lives for yeast cells and mammalian cells differ significantly. The cell types need to be specified when the authors state the range of the parameter.

7. For the NF-kappaB model, it is known that the outputs from the system are very noisy (e.g. https://doi.org/10.1038/s41467-025-60901-3), and only a small subset of the outputs have clear rhythmic patterns. How can the authors justify the importance of noise reduction and ultrasensitivity in this example?

8. MDM2 (R) is known to promote p53 degradation (A), and this is considered a crucial part of the negative feedback. This mechanism does not seem to be included in the authors’ model, so it is unclear whether it plays a role in noise reduction and whether its presence would affect the authors’ conclusion.

9. In this work, the authors seem to suggest that cooperative binding to multiple sites has no performance advantage compared to the indirect mechanisms in terms of noise reduction and ultrasensitivity maintenance. Given that many transcription repressors do bind to DNA at multiple sites, I suggest that the authors discuss why this mechanism is still positively selected in evolution.

10. Multisite binding without cooperativity can induce ultrasensitive responses when it is combined with other mechanisms, such as differential degradation rates from different complexes (doi: 10.1083/jcb.200308060, 10.15252/msb.20209945). These mechanisms may occur at protein or RNA levels, and some of them were shown to reduce noise. I suggest that the authors discuss these alternative mechanisms at different levels that may complement transcriptional regulations that the authors have focused on.

Reviewer #2: The paper by Jeong et al studies the effect of ultra-sensitive transcriptional responses in transcriptional noise. Authors build on their recent papers (references 18 and 19) and explore additional parameter regimes that are physiologically relevant and show that combination of indirect repression mechanisms of sequestration, blocking and displacement can achieve ulrasensitive responses that are associated to small transcriptional noise quantified by Fano factors. This paper is overall well-written and the results are ok. But, I have some concerns about the novelty and significance of the results presented.

Major comments:

- The advance over the resent work by authors (references 18, 19) seem minimal as it seems mostly an exploration of the new parameter regime. Also, the oscillation example and relevance to some biological systems (Figure 4) and the multiple target case (Fig. 5) is new. I invite the authors to make a better case on the advance developed in this paper.

- If I understand correctly, all the results both for the direct and indirect repressor assumes that the level of repressor is constant. But, in reality repressor is another protein and is being expressed stochastically and noice in the repressor can cascade through the transcriptional regulation to the down stream gene expression. This can be considered as a form of extrinsic noise. Is the proposed indirect repression mechanism also robust to this kind of noise?

- I was not completely clear what is done exactly with regards to the results in Figure 5. If the ratio of repressors to number of genes being regulated is close to 1 I would imagine the regulation becomes very noisy. I would like some more explanation about this. Also, this could be better presented as part of results rather than discussion.

Minor comments:

- Why is Figure 1a is not showing all possible states? I assumed that all states are allowed in the model. If this is just for illustration purposes should be explained in the caption.

- Table 1 bottom of page 10 in second column last line says "with the" seems like a word is missing.

- No code for the simulations has been included.

Reviewer #3: Ultrasensitive transcriptional response is a fundamental phenomenon in gene expression and gene regulation, playing a key role in shaping diverse biological behaviors. In this manuscript, the authors address an important and timely question: how can cells maintain low transcriptional noise while achieving highly ultrasensitive transcriptional responses under physiologically realistic conditions? Through careful modeling and analysis, the authors demonstrate that only the full combination of three indirect repression mechanisms—sequestration, blocking, and displacement—can consistently achieve both goals. The manuscript is well written, and the results are compelling and potentially impactful. I recommend publication after the following comments are adequately addressed.

Major points

1. The authors derive explicit analytical expressions for the transcriptional activities and Fano factors for three different models. It would strengthen the manuscript if the authors could further analyze these expressions and provide theoretical insights into why the combination of all three indirect repression mechanisms leads to low transcriptional noise, whereas combinations of only two do not. A mathematical or mechanistic explanation would significantly enhance the readers' understanding.

2. In Figure 1, the authors study ultrasensitivity generated by cooperative binding in the context of transcriptional repression. They show that this mechanism fails to simultaneously maintain low noise and high ultrasensitivity. Would the same conclusion hold if transcription were regulated by an activator protein instead of a repressor? A brief discussion or clarification of this scenario would be helpful for readers interested in activator-based regulation.

3. On page 6 and elsewhere, the authors present ranges for certain key parameters. However, such parameters can vary substantially across different cell types. It would be informative if the authors could specify the biological system or cell type that these parameter ranges are based on. Furthermore, providing a summary of typical Fano factor ranges observed across different cell types would give useful context to interpret the modeling results.

4. The mechanistic differences between Figure 4a–c and Figure 2a are not entirely clear. While the main text and captions suggest that these figures depict similar underlying mechanisms, there are apparent inconsistencies between them. For instance, "sequestration" is labeled under the gene-activator complex in Fig. 4a, while "blocking" is labeled under the same complex in Fig. 4b. Moreover, Fig. 4c lacks a gene-activator complex altogether. While the meaning of the solid arrows in these figures is clear, the roles of the dashed arrows and the solid or dashed lines without arrowheads are not immediately obvious. Clarifying the mechanistic distinctions behind these figures and the meanings represented by these graphical elements would greatly aid the reader's understanding.

Minor points

1. The authors have computed the transcriptional activities and Fano factors in closed form for several models. Could the authors also comment on whether other quantities—such as the distribution of mRNA counts and the autocorrelation function—can be derived in these models? These quantities provide further insights into stochastic gene expression. For instance: the time-dependent mRNA distribution has been derived for the model shown in Fig. 1a in:

-Analytical time-dependent distributions for gene expression models with complex promoter switching mechanisms. SIAM Journal on Applied Mathematics, 83(4):1572–1602, 2023,

and the autocorrelation function has been derived for a minimal negative feedback loop in:

-Exact power spectrum in a minimal hybrid model of stochastic gene expression oscillations. SIAM Journal on Applied Mathematics, 84(3):1204–1226, 2024.

A brief comment on the feasibility of similar calculations in the authors' framework would be valuable.

2. In the model that combines indirect repression with only sequestration, the following reactions may be missing in Table 1: "R+A \rightarrow R_A" and "R_A \rightarrow R+A".

3. Lines 156–158 are somewhat ambiguous in placement. It is unclear whether these lines are part of the caption for Table 1, the caption for Figure 1, or part of the main text. Please consider revising for clarity.

4. Line 174: "E_A" should likely be "E_R".

**Have the authors made all data and (if applicable) computational code underlying the findings in their manuscript fully available?**

Reviewer #1: Yes

Reviewer #2: **No: ** code for simulations is not included.

Reviewer #3: None

PLOS authors have the option to publish the peer review history of their article (what does this mean? ). If published, this will include your full peer review and any attached files.

**Do you want your identity to be public for this peer review?** For information about this choice, including consent withdrawal, please see our Privacy Policy .

Reviewer #1: No

Reviewer #2: No

Reviewer #3: No

**Figure resubmission:**

**Reproducibility:**



---

## [Decision Letter · Decision Letter 1]

3 Dec 2025

Dear Professor Kim,

We are pleased to inform you that your manuscript 'How Cells Tame Noise While Maintaining Ultrasensitive Transcriptional Responses' has been provisionally accepted for publication in PLOS Computational Biology.

Best regards,

Christopher E Miles

Academic Editor

PLOS Computational Biology

Mark Alber

Section Editor

PLOS Computational Biology

Reviewer's Responses to Questions

**Comments to the Authors:**

Reviewer #1: The authors have adequately addressed my comments with several new mechanistic insights. I'm recommending acceptance of this manuscript, and I congratulate the authors on this excellent work!

Reviewer #2: The authors have fully addressed my comments and have performed additional analysis to address them.

Reviewer #3: The authors have addressed all my previous comments satisfactorily. I strong recommend the paper for publication in PCB.

**Have the authors made all data and (if applicable) computational code underlying the findings in their manuscript fully available?**

Reviewer #1: Yes

Reviewer #2: Yes

Reviewer #3: None

PLOS authors have the option to publish the peer review history of their article (what does this mean? ). If published, this will include your full peer review and any attached files.

**Do you want your identity to be public for this peer review?** For information about this choice, including consent withdrawal, please see our Privacy Policy .

Reviewer #1: No

Reviewer #2: **Yes: ** Vahid Shahrezaei

Reviewer #3: **Yes: ** Chen Jia

---

## [Editor Report · Acceptance letter]

PCOMPBIOL-D-25-01132R1

How Cells Tame Noise While Maintaining Ultrasensitive Transcriptional Responses

Dear Dr Kim,

I am pleased to inform you that your manuscript has been formally accepted for publication in PLOS Computational Biology. Your manuscript is now with our production department and you will be notified of the publication date in due course.

With kind regards,

Aiswarya Satheesan
